

# From the XXZ chain to the integrable Rydberg-blockade ladder via non-invertible duality defects

Luisa Eck[1] and Paul Fendley[1,2]

**1** Rudolf Peierls Centre for Theoretical Physics, Parks Rd, Oxford OX1 3PU, United Kingdom
**2** All Souls College, Oxford, OX1 4AL, United Kingdom

## Abstract

Strongly interacting models often possess "dualities" subtler than a one-to-one mapping of energy levels. The maps can be non-invertible, as apparent in the canonical example of Kramers and Wannier. We analyse an algebraic structure common to the XXZ spin chain and three other models: Rydberg-blockade bosons with one particle per square of a ladder, a three-state antiferromagnet, and two Ising chains coupled in a zigzag fashion. The structure yields non-invertible maps between the four models while also guaranteeing all are integrable. We construct these maps explicitly utilising topological defects coming from fusion categories and the lattice version of the orbifold construction, and use them to give explicit conformal-field-theory partition functions describing their critical regions. The Rydberg and Ising ladders also possess interesting non-invertible symmetries, with the spontaneous breaking of one in the former resulting in an unusual ground-state degeneracy.

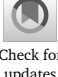

# 1 Intro

Statistical mechanics is full of "equivalences" and "dualities". Although both terms often are used rather loosely, the general idea is that different models are related by a clever rewriting of the degrees of freedom. One then expects that at minimum the physical properties of the models are related, and possibly identical. Kramers and Wannier's 1941 derivation of a "symmetry property" of the two-dimensional classical Ising model and its quantum-chain limit [1] remains a paradigm in the analysis of strongly interacting statistical mechanics. This property, typically known as "duality", is an exact linear relation between Ising partition functions at high and low temperature.

Kramers and Wannier's result was and remains subtle, as the low-temperature phase has two ground states (two minima of the free energy in the classical model), while the high-temperature phase has one. Thus "duality" is something of a misnomer, as becomes apparent after performing the transformation a second time. The result is not the original partition function, but rather that of a particular subsector. Kramers-Wannier "duality" therefore is a *non-invertible mapping* between the high- and low-temperature Ising models [2–5]. At a self-dual point where the map does not change the coupling, it is a *non-invertible symmetry*.

While the Kramers-Wannier mapping has been applied to other models, only recently has it been appreciated that it is a fundamental example of a large family of non-invertible mappings and symmetries. Starting with the pioneering construction of "topological symmetry" [6], a general structure has been developed to construct this family; see [7–9] and references therein. Fusion categories play a central role in this story, as they allow one to build models with a variety of nice properties. The Hamiltonian/transfer matrix can be written in terms of a set of generators obeying an algebra determined by the category. When two different models are both presentations of the same such algebra, their properties are at minimum related, and very possibly "equivalent".

One key property the category builds into the model is a set of *topological defects*. The partition function in the presence of such defects remains invariant under the deformation of

their paths, even if the defect separates regions described by different theories. Topological defects thus give a precise method for defining non-invertible mappings and symmetries. For example, the Ising duality defect away from the self-dual point separates a region in the ordered phase from a disordered region. Moving the defect around allows space to be filled with one coupling or the other and so relate the partition functions [5]. As a consequence of the non-invertibility, these relations are between linear combinations of partition functions with different boundary conditions and/or symmetry sectors. While much interesting recent work on such mappings has been devoted to the continuum versions (see [10] for an overview), our analysis is mainly on the lattice.

One advantage of the categorical approach is that close relations between seemingly different models are quickly revealed. The Temperley-Lieb algebra [11] provides a classic example. Originally used to prove an "equivalence" between the $Q$-state Potts models and the six-vertex model/XXZ chain, it goes much deeper. Its graphical presentation in terms of clusters or the surrounding self-avoiding loops [12, 13] made it central to Jones's construction of knot and link invariants [14]. The development of fusion categories was spurred on by this connection to statistical mechanics [15, 16], along with closely related developments in conformal field theory [17].

Our starting point is the XXZ chain, but we follow Temperley and Lieb only in spirit. We write its Hamiltonian in terms of a set of generators satisfying a distinct algebra, given in (1, 2) below. This algebra has a nice graphical presentation arising from the *chromatic algebra* [18, 19]. The latter algebra was developed to give a systematic approach to computing identities for a famed topological invariant, the chromatic polynomial. It is similarly useful here, allowing us to relate the XXZ chain to a three-state antiferromagnetic chain. It also allows us put the XXZ chain in a categorical setting distinct from the Temperley-Lieb approach [7, 20]. This setting leads to our construction of topological defects giving exact non-invertible mappings from XXZ and the antiferromagnet to two quantum ladders. One ladder describes bosons obeying a Rydberg blockade, and the other two Ising models coupled in a zigzag fashion.

The purpose of this paper is to define and analyse these four models explicitly and precisely, including a careful treatment of the non-invertible mappings and symmetries. While many of the connections between them were explored before using integrability [21–23], anyon chains [24, 25] and categories [7, 20], our approach unifies the various threads. For example, the algebra we describe yields a simple and direct proof that all four models are integrable. Being explicit clarifies many of the subtle aspects of these mappings, and so we hope this paper serves as a useful guide to the subject. The models and the mappings between them have a virtue of being reasonably transparent in their properties, showing that Ising is not the only basic example.

Moreover, the models are very interesting in their own right. Rydberg-blockade bosons in particular have been the subject of intense study because of their experimental accessibility [26], striking theoretical properties such as quantum scars [27], and fascinating proposals for realising spin-liquid phases [28, 29]. The integrable Rydberg-blockade ladder we describe provides one of the few exact theoretical results for such models outside of the chain. As such, in our companion paper [30], we use these results to explore the phase diagram away from the integrable line.

The key underlying algebra and its properties are analysed in section 2. We give several graphical presentations, including one arising from the chromatic algebra and another from the fusion category $su(2)_4$. Using guidance from the category, four integrable models satisfying this algebra are found in section 3. We generally describe them in terms of their quantum Hamiltonians, but analogous two-dimensional classical models are constructed as well. A further payoff of the category approach comes in section 4, where we provide explicit and exact non-invertible mappings between all four models. We derive a number of simple relations be-

tween them, enabling us to determine in which sectors a given map is non-trivial. This analysis is extended to include modified maps acting on "twisted" sectors, enabling the full spectra of the other models to be related to that of the XXZ chain with appropriately chosen boundary conditions. In section 5, we go further and show how two models have non-invertible symmetries. One striking result is the existence of a gapped line in the Rydberg ladder with a ground-state degeneracy arising from the spontaneous breaking of a non-invertible symmetry. In section 6 we use our mappings and symmetries to analyse the critical region, relating the spectrum of the other three models to that of the XXZ chain. We find exact partition functions for all models in the continuum limit, using the orbifold technique from conformal field theory [31, 32] and the lattice [33], showing for example how the integrable Rydberg ladder is an $S_3$ orbifold of the XXZ chain. In appendix A we give the explicit MPO form of one of the maps, while in appendix B we review the mapping of the XXZ chain to the free-boson conformal field theory.

## 2 The algebra and its presentations

The classic result of Temperley and Lieb expresses the XXZ Hamiltonian in terms of the generators of an algebra bearing their name [11]. The transfer matrices/Hamiltonians of other seemingly different models have the same expressions when written in terms of these generators. The models thus are closely related, and it is often said that they are "equivalent". However, to make the correspondences precise, one must analyse the symmetry sectors and the corresponding boundary conditions. In this paper, we utilise a different algebra from Temperley and Lieb to construct distinct models related by non-invertible mappings. In this section, we give this algebra and show how it has appeared in a number of guises, including several graphical presentations.

### 2.1 The algebra

All the Hamiltonians we study are expressed in terms of two sets of generators $S_j$ and $P_j$, with $j = 1, 2, \ldots L$. The Hamiltonian is local, as

$$S_j P_k = P_k S_j, \quad S_j S_k = S_k S_j, \quad P_j P_k = P_k P_j, \text{ for } |j - k| \neq 1. \tag{1}$$

The indices are interpreted cyclically so that e.g. $S_1$ and $S_L$ do not commute. The interesting part of the algebra is

$$\boxed{\left(S_j\right)^2 = 1 - P_j, \quad \left(P_j\right)^2 = P_j, \quad S_j P_j = 0, \quad S_j S_{j\pm1} S_j = P_j S_{j\pm1} P_j = 0.} \tag{2}$$

The Hamiltonians depend on a single parameter $\Delta$:

$$H = \sum_{j=1}^{L} \left(S_j + \Delta P_j\right). \tag{3}$$

The cyclicity of the indices results in periodic boundary conditions in the models we study.

The XXZ model is the best-known presentation of the algebra (1, 2) [34, 35]. Here the Hilbert space $\mathcal{H}_{\mathrm{XXZ}}$ is a chain of $L$ two-state systems, i.e. $(\mathbb{C}^2)^{\otimes L}$. The operators $X_{j+\frac{1}{2}}$, $Y_{j+\frac{1}{2}}$ and $Z_{j+\frac{1}{2}}$ act as Pauli matrices on the system labeled by $j + \frac{1}{2}$ for integer $j$, and trivially elsewhere, i.e. are of the form $1 \otimes 1 \otimes \cdots X \cdots \otimes 1$. The XXZ Hamiltonian with periodic boundary conditions is

$$H_{\mathrm{XXZ}} = \frac{1}{2} \sum_{j=1}^{L} \left(X_{j-\frac{1}{2}} X_{j+\frac{1}{2}} + Y_{j-\frac{1}{2}} Y_{j+\frac{1}{2}} + \Delta(1 + Z_{j-\frac{1}{2}} Z_{j+\frac{1}{2}})\right). \tag{4}$$

Antiferromagnetic and ferromagnetic interactions between adjacent spins correspond to setting the coupling $\Delta > 0$ and $\Delta < 0$ respectively. Quite obviously, the XXZ Hamiltonian (4) is of the form (3) with

$$S_j = \tfrac{1}{2}\Big(X_{j-\frac{1}{2}}X_{j+\frac{1}{2}} + Y_{j-\frac{1}{2}}Y_{j+\frac{1}{2}}\Big), \qquad P_j = \tfrac{1}{2}\Big(1 + Z_{j-\frac{1}{2}}Z_{j+\frac{1}{2}}\Big). \tag{5}$$

Less obvious, but easy to show, is that these definitions yield a presentation of the algebra (1, 2). One key difference between our and the usual Temperley-Lieb approach to the XXZ chain is that in the latter, the generators explicitly incorporate the parameter $\Delta$. Here and everywhere in this paper, $\Delta$ is simply a parameter in the Hamiltonian and is not involved in the algebra.

The relations (2) immediately imply a host of other relations, the simplest being

$$P_j S_j = 0, \qquad P_j P_{j+1} = P_{j+1} P_j, \qquad P_j S_{j\pm 1} = S_{j\pm 1}\big(1 - P_j\big). \tag{6}$$

The first follows from $P_j S_j = S_j - (S_j)^3 = S_j P_j = 0$. The latter follows from

$$0 = S_j\big(S_j S_{j\pm 1} S_j\big)S_j = (1 - P_j)S_{j\pm 1}(1 - P_j) = S_{j\pm 1} - S_{j\pm 1} P_j - P_j S_{j\pm 1}. \tag{7}$$

It follows that in the definition of the algebra, the relation $P_j S_{j\pm 1} P_j = 0$ could be replaced by the latter of (6). The fact that all $P_j$ commute follows from comparing

$$P_j P_{j+1}\big(1 - P_j\big) = P_j P_{j+1}\big(S_j\big)^2 = P_j\big(S_j\big(1 - P_{j+1}\big)\big)S_j = 0 \quad \Longrightarrow \quad P_j P_{j+1} P_j = P_j P_{j+1},$$

$$\big(1 - P_j\big)P_{j+1}P_j = \big(S_j\big)^2 P_{j+1}P_j = P_j\big(\big(1 - P_{j+1}\big)S_j\big)P_j = 0 \quad \Longrightarrow \quad P_j P_{j+1} P_j = P_{j+1} P_j.$$

## 2.2 Graphical presentation from the chromatic algebra

In the rest of this section we explain how the algebra (1, 2) used to define $H$ is a very special case of several known algebras. Here we show how the "chrome-plus" algebra, a particular chromatic algebra extended with an extra relation, yields (1, 2). In section 2.3, we prove the converse, showing that (1, 2) define a certain BMW algebra, which in turn is equivalent to the chrome-plus algebra. In section 2.4 we place these results in the more general setting of fusion categories.

The chromatic algebra relations are linear identities between distinct graphs that allow one to "evaluate" any planar graph, i.e. associate a number with each graph depending only on its topological properties [18, 19]. This topological invariant, the chromatic polynomial, is invariant under all continuous deformations of the graphs. Before defining the chromatic polynomial, we discuss these identities. We need consider here only planar trivalent graphs with no ends (i.e. no 1-valent vertices). Evaluating such graphs requires three identities [18, 19], each of which holds for any subgraph. The simplest are that removing a closed detached loop gives a factor of $Q - 1$, and that the evaluation of any graph with a tadpole vanishes:

$$\bigcirc = (Q - 1), \qquad \multimap\!\!\bigcirc = 0. \tag{8}$$

The third identity arises from the contraction-deletion property of the chromatic polynomial. In terms of these graphs, it is

$$\asymp + \underset{\smile}{\frown} = \rightangle\!\!\langle + \,)\,(\,. \tag{9}$$

Whenever any of the graphs in (8) and (9) appears as a subgraph, we can replace it with the respective number or linear combination without changing the evaluation.

Repeatedly applying (9) allows any planar trivalent graph with no ends to be turned into a sum over collections of loops and tadpoles, which then can be evaluated using (8). For example, using all three of the identities gives

$$\text{———} = \text{———} + \text{———} - \text{———} = (Q-2)\ \text{———}\ . \qquad (10)$$

This identity for bubble removal then can be used to evaluate for example

$$\bigodot = (Q-2)\bigcirc = (Q-2)(Q-1)\,. \qquad (11)$$

A useful one in our analysis is triangle removal:

$$\text{———} = \text{———} + \text{———} - \text{———} = (Q-3)\ \text{———}\ . \qquad (12)$$

The proof is simple: the first graph in the sum simplifies using (10) (recall all relations hold under rotations), the second vanishes because of (8), and the third already is in the desired form.

The evaluation here is called the chromatic polynomial. This toplogical invariant has a long history, as it gives a way to generalise the notion of $Q$-colouring to any $Q$. Namely, treat the lines in a planar graph as being the boundaries of regions in a map (in the colloquial sense of the word "map"). A $Q$-colouring is a way of assigning one of $Q$ colours to each of the regions, such that any two regions separated by an edge must be coloured differently. The chromatic polynomial $\chi_{\hat{\mathcal{G}}}(Q)$ is the number of ways of colouring the faces of a planar graph $\mathcal{G}$ (we label by the dual graph $\hat{\mathcal{G}}$ to conform to the usual convention). The famed four-colour theorem says that $\chi_{\hat{\mathcal{G}}}(4) > 0$ for all $\mathcal{G}$.

The correspondence between the chromatic algebra and the chromatic polynomial is simple and direct: Using rules (8,9) yields $\chi_{\hat{\mathcal{T}}}(Q)/Q$ for the evaluation of any planar trivalent graph $\mathcal{T}$ [18,19]. Since $Q$ is only a parameter in these rules, this procedure gives a definition of the chromatic polynomial for all $Q$. The evaluation in terms of colourings is already apparent in the examples above. Consider the graph $\mathcal{L}$ comprised simply of a single loop embedded in the plane. The two regions must be coloured differently, so the number of ways of colouring the two regions is $\chi_{\hat{\mathcal{L}}}(Q) = Q(Q-1)$. Using the first rule in (8) means that the evaluation is simply $Q-1$, indeed equal to $\chi_{\hat{\mathcal{L}}}(Q)/Q$. Similarly, the three regions in (11) each need to be coloured differently, so the corresponding chromatic polynomial is $Q(Q-1)(Q-2)$, in agreement with the graph evaluation times $Q$. Likewise, the identity in (12) follows from noting that all four regions in the picture on the left must be coloured differently.

The operators obeying the algebra (1, 2) can be presented in terms of planar trivalent graphs obeying the chromatic algebra [18, 19], plus one more relation we describe below. The operators in this graphical presentation are defined to act on a set of $L$ vertical strands by causing them to branch and join. Specifically, the $P_j$, $S_j$ and a third kind of generator $E_j$ are given by

$$P_j = \ \times\ , \qquad S_j = \ \rangle\langle\ , \qquad E_j = \ \asymp\ , \qquad (13)$$

where a generator labelled by $j$ acts on the $j$ and $j+1$ strands (interpreted as always cyclically). The contraction-deletion relation (9) of the chromatic algebra gives a linear relation between these generators, namely

$$P_j + E_j = 1 + S_j\,. \qquad (14)$$

To see how (13) and the chromatic algebra yield the algebra (1, 2), it is first instructive to look at the generators $E_j$. From the point of view of the latter algebra, the contraction-deletion relation provides the definition of the $E_j$. From the point of view of the chromatic algebra, the subalgebra involving only the generators $E_j$ is the Temperley-Lieb algebra [11], in its graphical presentation [12, 13]. Using the chromatic algebra to get rid of the closed loop means that

$$(E_j)^2 = \quad = (Q-1) \quad = (Q-1)E_j, \qquad E_j E_{j+1} E_j = \quad = \quad = E_j, \tag{15}$$

along with $E_j E_k = E_k E_j$ for $|j - k| > 1$. The generators $E_j$ are related to the $S_j$ and $P_j$ via (14), and so if (2) is to be satisfied, then

$$E_j^2 = (S_j + 1 - P_j)^2 = 2(S_j + 1 - P_j) = 2E_j. \tag{16}$$

Comparing this relation to the graphical one (15) means that the correspondence between algebras is possible only when the number of colours is $Q = 3$.

We now can see how far we can get using just the chromatic algebra. Obviously, operators with labels $j$ and $j'$ commute with $|j - j'| > 1$, yielding (2). The first line of (1) follows from the above relations in the chromatic algebra. For example,

$$S_j P_j = \quad = 0, \qquad (P_j)^2 = \quad = \quad = P_j, \tag{17}$$

follow respectively from (12) and (10) with $Q = 3$. Showing $(S_j)^2 = 1 - P_j$ takes a few steps, but is straightforward using the above chromatic identities. The relation

$$P_j S_{j+1} P_j = \quad = 0, \tag{18}$$

follows from (12).

However, the remaining relation in (2) does not follow solely from (8) and (9). Instead, we must exploit an additional property special to $Q = 3$. The key observation is that when $q$ defined by $Q = (q^{\frac{1}{2}} + q^{-\frac{1}{2}})^2$ is a root of unity, the evaluations of graphs using the chromatic algebra obey more linear identities [18]. It then becomes consistent to set one particular Jones-Wenzl projector [36] to be zero, and so enhance the chromatic algebra accordingly. The procedure for finding this projector is reviewed in detail in [18], so we simply present the result here. For $Q = 3$, we have

$$= 0. \tag{19}$$

This identity is consistent with the fact that the evaluation is the chromatic polynomial. Indeed, since $Q$ is an integer, we can use colourings. This interpretation shows immediately why the graph in (12) vanishes at $Q = 3$, because colouring all four regions requires four distinct colours. Similarly, the Jones-Wenzl projector in (19) cannot be coloured with three colours:

one colour is needed for inside the circle, while the other two colours must then alternate in the regions outside the circle. Since there are an odd number of the latter regions, such a colouring is impossible and the evaluation must vanish. Extending the $Q = 3$ chromatic algebra by (19) immediately yields the remaining relation in (2):

$$S_j S_{j+1} S_j = \quad \boxed{} \quad = 0. \tag{20}$$

We thus have proved that the algebra (1, 2) follows from the "chrome-plus" algebra at $Q = 3$. The latter algebra consists of the chromatic algebra enhanced by the Jones-Wenzl projector (19). The converse also holds, which we prove next.

## 2.3 Graphical presentation in terms of the BMW algebra

Pioneering work in the 70s and 80s showed the close connection between algebras, integrable lattice models and knot and link invariants [11–15]. The example we study is no exception, and indeed, has quite an elegant description in the knot and link language. We exploit these results to prove the converse of the result of the previous section. Namely, we show that the algebra (1, 2) yields a special case of the $so(3)$ BMW algebra [37, 38] enhanced by its Jones-Wenzl projector [18, 36]. Since the latter is equivalent to the chrome-plus algebra [19, 39], we thus have proved that the two are completely equivalent.

As with the chromatic algebra, the BMW algebra has a beautiful graphical presentation. It includes over- and under-crossings, thus allowing one to compute knot and link invariants generalising the Jones polynomial. The BMW algebras also provide centralizers of quantum-group algebras [40], and can be used to construct lattice models invariant under the latter. For example, the integrable Fateev-Zamolodchikov spin-1 chain [41] is invariant under the quantum group $U_q(\mathfrak{so}_3)$, and the analogous RSOS height models come from projecting on to singlets under the symmetry [42]. We will in essence follow this procedure to construct lattice Hamiltonians below, albeit in the equivalent fusion-category language.

Here we show how to obtain the BMW algebra from (1, 2). The simplest part of the connection is to construct a representation of the braid group. The generators are

$$B_j = q P_j + q^{-1} S_j, \qquad B_j^{-1} = q^{-1} P_j + q S_j, \tag{21}$$

for some parameter $q$. Using (2) and the ensuing identities such as (6) gives

$$B_j B_{j+1} B_j = q^3 P_j P_{j+1} + q^{-1} \big( S_j S_{j+1} + S_{j+1} S_j + \big(1 - P_j\big)\big(1 - P_{j+1}\big)\big).$$

Since $P_j$ and $P_{j+1}$ commute, this expression is invariant under exchanging $j \leftrightarrow j+1$. Therefore it also equals $B_{j+1} B_j B_{j+1}$, showing that the $B_j$ indeed satisfy the braid-group relation

$$B_j B_{j+1} B_j = B_{j+1} B_j B_{j+1}. \tag{22}$$

In the graphical presentation of the braid group, these generators act on a set of $L$ vertical lines, i.e. on a vector space of all possible connections of $L$ points at the bottom with $L$ points at the top. Then $B_j$ corresponds to an overcrossing of $j$ and $j + 1$th lines (with index interpreted cyclically as always), and $B_j^{-1}$ an undercrossing. These and the Temperley-Lieb generator $E_j$ are pictured as

$$B_j = \quad \diagdown\!\!\!\!\diagup \quad, \qquad B_j^{-1} = \quad \diagup\!\!\!\!\diagdown \quad, \qquad E_j = \quad \smile\!\!\smile \quad. \tag{23}$$

Multiplication is by stacking, and then one can derive identities by moving the lines around in three-dimensional space. For example, the second Reidemeister move is simply

$$B_j B_j^{-1} = \quad = \quad = 1 \,, \tag{24}$$

while the braid-group relation (22) is the third Reidemeister move:

$$B_j B_{j+1} B_j = \quad = \quad = B_{j+1} B_j B_{j+1} \,. \tag{25}$$

The graphical proof of the latter is simply dragging the middle line horizontally.

The BMW algebra enhances the braid-group relations to include the Temperley-Lieb generator $E_j$. A variety of relations among the generators ensure that topological invariants of knots and links can be constructed. An explicit list can be found in [37]; we follow the conventions of [39, 43]. We then can prove that (1, 2) implies the BMW algebra and hence the chrome-plus algebra simply by going through them one-by-one. For each $j$ there is a linear relation among the three generators, which allows us to fix conventions: comparing (14) and (21) means that

$$B_j - B_j^{-1} = \left( q - q^{-1} \right) \left( 1 - E_j \right). \tag{26}$$

In the knot/link context, this relation is often called the Kaufmann skein relation.

The relations involving two generators with the same $j$ also follow easily. We have already used the fact that $E_j^2 = 2 E_j$ follows from (2) to fix $Q = 3$. The definition (21) then yields

$$B_j E_j = (q P_j + q^{-1} S_j)(1 - P_j + S_j) = q^{-1} \left( S_j + 1 - P_j \right) = q^{-1} E_j \,. \tag{27}$$

These allow us to make contact with the first Reidemeister move in the graphical presentation:

$$B_j E_j = \quad = q^{-1} \quad = q^{-1} E_j \,. \tag{28}$$

Similarly, $B_j^{-1} E_j = E_j B_j^{-1} = q^{-1} E_j$, so that a clockwise twist results in a factor $q$, and a counterclockwise twist a factor $q^{-1}$. These extra factors mean this approach does not quite give the first Reidemeister move. To construct a knot or link invariant, one must keep track of these twistings to remove the resulting factors of $q$. This is best done by framing the knot, i.e. turning each strand into a ribbon, and then computing the resulting "writhe" [14].

The remaining BMW relations involve generators on three strands. We have already showed that the braid-group relation (22) holds. Another one is

$$E_j B_{j+1} E_j = \left( 1 - P_j + S_j \right) \left( q P_{j+1} + q^{-1} S_{j+1} \right) \left( 1 - P_j + S_j \right) = q \left( 1 - P_j + S_j \right) = q E_j \,, \tag{29}$$

again following from (2) and (6). The factor of $q$ is needed for the correct twisting:

$$E_j B_{j+1} E_j = \quad = q \quad = q E_j \,.$$

Of the remaining identities, one is the Temperley-Lieb relation $E_j E_{j+1} E_j = E_j$ described above. The others are [37, 43]

$$B_j B_{j+1} E_j = E_{j+1} E_j, \qquad B_j E_{j+1} B_j = B_{j+1}^{-1} E_j B_{j+1}^{-1}, \qquad B_j E_{j+1} E_j = B_{j+1}^{-1} E_j, \qquad (30)$$

along with their vertical and horizontal reflections. All have nice interpretations as graphical identities found by dragging lines around, as with the braid-group relation. It is straightforward to show that with the relation (21), all are satisfied by using manipulations using the algebra (2).

We have not yet identified which BMW algebra this is, and have not addressed the Jones-Wenzl projector. These two issues are connected. Conventionally, $so(N)$ BMW algebras are labelled by two parameters $N$ and $k$, with the associated quantum-group parameter defined as $q = e^{i\pi/(k+N-2)}$. The representations are most interesting when $N$ and $k$ are integers, so that $q$ is a root of unity. To be the centralizer of the quantum-group algebra $U_q(\mathfrak{so}(N))$, the simplest choice for the twist factor is $q^{N-1}$, with the above conventions [37, 43]. We thus may identify $N = 2$. The parameter $q$ and the parameter $k$ then are not constrained so far: all the BMW relations above hold for any $q$. This identification should not come as a shock, as the XXZ chain has a $SO(2) = U(1)$ symmetry. However, this identification is not unique, because these BMW algebras have a level-rank duality exchanging $N$ with $k$. The point is that because $q^{N-1} = -q^{1-k}$, all the BMW relations hold under the substitution $B \leftrightarrow -B^{-1}$ and $N \leftrightarrow k$. Thus we equally could have identified the corresponding BMW algebra as having $k = 2$ and $N$ as of yet undetermined.

Including the Jones-Wenzl projector does fix the value of $q$ and hence both $N$ and $k$. The reason is that only for a specific value of $q$ is imposing this constraint consistent with the other relations. We already saw this property in connection with the chromatic algebra: only for $Q = 3$ is it consistent to impose (19) or equivalently $S_j S_{j+1} S_j = 0$. A quick way to identify $q$ here is by the relation $E_j^2 = d_1 E_j$, as the expressions for $d_1$ are known in both the chromatic and the BMW algebras, giving [19, 39]

$$d_1 = Q - 1 = 1 + \frac{q^{N-1} - q^{1-N}}{q - q^{-1}}. \qquad (31)$$

Setting $N = 2$ does indeed yield $Q = 3$ for any $q$. However if we exploit the level-rank duality and instead fix $k = 2$, then imposing $Q = 3$ requires that $N = 3$, and $q = e^{i\pi/3}$. The BMW algebra relevant here thus can be labelled as $so(3)_2$, and is the centralizer of the quantum-group algebra $U_q(\mathfrak{so}(3))$. This labelling is in harmony with the correspondence between the chromatic and BMW algebras, which in general requires $N = 3$ in the latter. The general relation between $Q$ of the chromatic algebra and $q$ of the BMW algebra is then $Q = (q^{\frac{1}{2}} + q^{-\frac{1}{2}})^2$.

## 2.4 Fusion categories

We have shown that (1) and (2) is equivalent to the $so(3)_2$ BMW algebra, which in turn is equivalent to the chrome-plus algebra. This structure goes deeper, as this algebra forms part of a *fusion category*. Fusion categories allow one to construct isotopy invariants of labelled graphs, and are familiar in physics for understanding the topological properties of anyons. As with the chromatic algebra, a nice feature of the fusion category is that it allows one to compute isotopy invariants of planar graphs without needing to discuss the specific presentation acting on vertical strands. Many fine reviews of fusion categories from a variety of perspectives already exist, such as [17, 44], and in particular all the relevant information for the models described here can be found in [7]. We thus mainly focus on the case at hand.

A fusion category subsumes the algebras we have discussed, giving a more general structure. The basic data for a fusion is a list of objects obeying *fusion rules* describing the tensor

product of the objects. In an equation,

$$a \otimes b = \sum_c N_{ab}^c c, \tag{32}$$

where the $N_{ab}^c$ are non-negative integers. An example of such a set are certain irreducible representations of quantum-group algebras with $q$ a root of unity (ordinary Lie algebras obey the same kind of fusion rules, but the number of representations and hence objects is not finite). The category we study has some simplifying properties: fusion is associative, and all $N_{ab}^c$ take values of 0 or 1 and remain invariant under permutations of $a$, $b$ and $c$.

The chromatic and BMW algebras in general are not automatically embeddable into fusion categories, as the associated fusion algebras generically involve an infinite number of objects. However, when $q$ is a root of unity one can impose the Jones-Wenzl projector and make the number of objects finite. The chrome-plus (i.e. $N = 3$, $k = 2$ BMW) algebra analysed here extends to a category usually known as $so(3)_2$. In this case there are three objects, which are labelled as 0, 1, and 2 in the standard physics conventions for labelling spins (they correspond to labels 0, 2 and 4 in [18, 19]). The object labeled by 0 is the identity, so that the fusion $s \otimes 0 = s$ for any $s$, and lines in a graph labelled by 0 can simply be omitted. The other fusion rules of $so(3)_2$ are

$$1 \otimes 1 = 0 \oplus 1 \oplus 2, \qquad 2 \otimes 1 = 1, \qquad 2 \otimes 2 = 0. \tag{33}$$

The 0 in the last relation means the object labelled by 0, not the number; two objects always fuse to something non-trivial. It is not coincidental that the fusion rule of the object 1 resembles that of a spin-1 representation of $so(3)$ or $su(2)$: it survives under the truncation resulting from setting $k = 2$. Indeed, the fusion category $so(3)_2$ is the subcategory comprised of the integer-spin objects of $su(2)_4$, which is associated with the quantum group $U_q(\mathfrak{sl}_2)$. The larger category $su(2)_4$ (and its close relative $\mathcal{A}_5$) have objects labelled by spins, 0, $\frac{1}{2}$, 1, $\frac{3}{2}$, and 2. The fusion rules are

$$
\begin{aligned}
s \otimes 0 = s, \qquad s \otimes 2 = (2-s), \qquad 1 \otimes 1 = 0 \oplus 1 \oplus 2, \\
\tfrac{1}{2} \otimes 1 = \tfrac{3}{2} \otimes 1 = \tfrac{1}{2} \oplus \tfrac{3}{2}, \qquad \tfrac{1}{2} \otimes \tfrac{1}{2} = 0 \oplus 1, \qquad \tfrac{1}{2} \otimes \tfrac{3}{2} = 1 \oplus 2,
\end{aligned}
\tag{34}
$$

for any $s$.

Just as with the chromatic algebra, a fusion category contains a list of rules that allow one to evaluate a planar trivalent graph, but where each edge is labelled by one of the objects. When an object $c \in a \otimes b$ (i.e. $N_{ab}^c \geq 1$), then the three edges around a trivalent vertex can be labelled by $a$, $b$ and $c$. Because of the simpler nature of our categories, the edges are unoriented, and the labels can be placed in any order. The simplest rules are in (8): tadpoles vanish no matter what the labels, while a closed loop labeled by $a$ receives a weight $d_a$, its "quantum dimension". In addition to the quantum dimensions, the other data provided by the fusion category are the coefficients under "$F$ moves". These moves give a linear relation between the two ways four strands can fuse together in a planar trivalent graph. In $so(3)_2$, the fusion rule $1 \otimes 1 = 0 \oplus 1 \oplus 2$ means that for $s = 0, 1, 2$ there exist $F$ moves of the form

$$
\left\langle \!\! \begin{array}{c} s \end{array} \!\! \right\rangle = \sum_{r=0,1,2} f_r^{(s)} \; \right\rangle\!\! r \!\!\left\langle \; = f_0^{(s)} \; \smile\!\!\frown \; + f_1^{(s)} \; \rangle\!\langle \; + f_2^{(s)} \; \rangle\!\vdots\!\langle \, , \tag{35}
$$

where an unlabelled solid line corresponds to spin-1 and the dotted to spin-2, and any line labelled by 0 can be omitted. The coefficients $f_r^{(s)}$ are called the $F$ symbols, and they must satisfy a variety of constraints such as the pentagon equation in order to make the evaluation unique.

Using $F$ moves and the other rules allow one to replace two concentric loops with a single one, as with the fusion rules (32). This in turn puts a constraint on the quantum dimensions:

$$\left( \bigcirc a \right) b \; = \; \sum_c N^c_{ab} \; \bigcirc c \qquad \implies \qquad d_a d_b = \sum_c N^c_{ab} d_c \,. \tag{36}$$

The identity defect must always have $d_0 = 1$, so in $so(3)_2$ this relation along with fusion (33) gives $d_1 = 2$, $d_2 = 1$ (quantum dimensions must always be positive in the unitary categories we consider). The quantum dimensions of the half-integer-spin objects in $su(2)_4$ are then $d_{\frac{1}{2}} = d_{\frac{3}{2}} = \sqrt{3}$.

The extension of the chrome-plus algebra to the $so(3)_2$ fusion category is not immediately obvious, as the graphs in the latter have labels where those for the former do not. The reason for the seeming discrepancy is straightforward to understand: one can use the $F$ moves to remove all occurrences of the label 2, leaving only lines with label 1. Namely, the $F$ moves give three linear relations among 6 graphs with four external spin-1 legs. They can then be used to relate the two graphs with spin-2 lines to a sum over those without it. The remaining spin-1 lines are described by the chromatic algebra. A convenient way of making the correspondence precise is to identify the projectors/idempotents acting on strands. There are three sets of projectors $P_j^{(0)}$, $P_j^{(1)}$, $P_j^{(2)}$ acting on two spin-1 strands, obeying

$$P_j^{(a)} P_j^{(b)} = \delta_{ab} P_j^{(a)}\,, \qquad P_j^{(0)} + P_j^{(1)} + P_j^{(2)} = 1\,, \tag{37}$$

for all $j$. Graphically, these projectors take two spin-1 lines and fuse them to a single line, as on the right-hand side of (35). Then in the chromatic algebra

$$P_j^{(0)} = \tfrac{1}{2} E_j = \tfrac{1}{2} \smile\frown \,, \qquad\qquad P_j^{(1)} = P_j = \; \times \qquad . \tag{38}$$

The graph for $P^{(2)}$ then corresponds to having the vertical line labelled by 2 (i.e. the dotted line in (35)), but by construction we must have $P_j^{(2)} = 1 - P_j^{(0)} - P_j^{(1)} = 1 - \tfrac{1}{2} E_j - P_j$ here. This relation thus fixes the $F$ symbols in (35) with $s = 0$. (Our normalisation of the $F$ symbols is not the conventional one, but rather the one set by the chromatic algebra. The conventional $F$ symbols for $so(3)_k$ can be found in equation (3.18) of [7], and the evaluation rules change correspondingly by a factor of $2^{1/4}$ for each trivalent vertex of spin-1 lines.) We then have $S_j = P_j^{(0)} - P_j^{(2)}$.

Since all our relations remain true under rotations, the horizontal spin-2 line also can be replaced by linear combinations of graphs involving only the spin-1 lines, i.e. the unlabelled ones in the chromatic algebra. Replacing the spin-2 lines requires using two of the three identities in (35). The third thus results in an additional constraint: it is precisely the contraction-deletion relation (9) of the chromatic algebra! Thus the graphical relations in the $so(3)_2$ fusion category are those of our chrome-plus algebra. Moreover, this category can be extended to have braiding, yielding that described in section 2.3.

# 3 A quartet of Hamiltonians

In the previous section we gave a variety of presentations of the algebra (1, 2), and showed how these presentations arise in the $so(3)_2$ and $su(2)_4$ fusion categories. Here we exploit these connections to display different integrable Hamiltonians satisfying this algebra.

### 3.1 Integrability

As has long been known, the XXZ chain is integrable and solvable by the Bethe Ansatz [45]. One way of proving its integrability is to show that the Boltzmann weights of its classical 2d generalisation, the six-vertex model, obey the Yang-Baxter equation (YBE). The XXZ chain is then obtained by taking a special limit of these Boltzmann weights. The YBE guarantees that the ensuing Hamiltonian commutes with the transfer matrix of the classical model, yielding a hierarchy of commuting charges that imply integrability.

One does not need to use the XXZ presentation to prove integrability, however: Boltzmann weights from the algebra (1, 2) is sufficient to guarantee a solution of the YBE [34, 35]. Any Hamiltonian of the form (3), including the three other ones described in this section, is thus integrable. The Yang-Baxter equation in difference form is

$$R_j(u)R_{j+1}(u+u')R_j(u') = R_{j+1}(u')R_j(u+u')R_{j+1}(u), \tag{39}$$

where $u$ is called the spectral parameter. The resemblance to the braid-group relation (22) is not a coincidence, and using a braided tensor category to find a full $u$-dependent solution to the YBE is called "Baxterisation" [46].

We parametrise this solution by

$$R_j(u) = 1 - P_j + \alpha(u)S_j + \beta(u)P_j. \tag{40}$$

Demanding $R$ of this form satisfy (39) and using (2) and (6) yields two functional equations for $\alpha(u)$ and $\beta(u)$:

$$\beta(u+u') + \alpha(u)\alpha(u') = \beta(u)\beta(u'), \qquad \alpha(u) + \beta(u+u')\alpha(u') = \beta(u)\alpha(u+u'). \tag{41}$$

The relations (41) are solved by $\alpha(u) = \sin u/\sin \mu$ and $\beta(u) = \sin(u+\mu)/\sin \mu$

$$R_j(u) = 1 - P_j + \frac{\sin u}{\sin \mu}S_j + \frac{\sin(u+\mu)}{\sin \mu}P_j, \tag{42}$$

satisfies the YBE for any value of the parameter $\mu$. Worth noting is that the operator $R_j(u)$ is invertible for $u \neq \pm\mu$:

$$R_j(u)R_j(-u) = \frac{\sin(\mu+u)\sin(\mu-u)}{\sin^2 \mu}. \tag{43}$$

The second and third Reidemeister moves follow from (43) and (39) by taking $B_j \propto R_j(i\infty)$.

A transfer matrix of an integrable classical lattice model then can be constructed by taking the product of all these operators as

$$T(u) = R_L(u)R_{L-1}(u)\ldots R_2(u)R_1(u). \tag{44}$$

The YBE then requires that $[T(u), T(u')] = 0$ for any $u$ and $u'$ [45]. The quantum Hamiltonian (3) then follows from taking the $u \to 0$ limit, giving

$$T(u) = 1 + \frac{u}{\sin \mu}H + \mathcal{O}(u^2), \quad \text{where} \quad \Delta = \cos \mu. \tag{45}$$

Expanding the commutator in powers of $u'$ gives $[H, T(u)] = 0$ for any $\Delta$, since the YBE holds for any $\mu$. Expanding $d\ln(T(u))/du$ in $u$ gives a series of local conserved quantities, so any $H$ built by this approach is integrable for any $\Delta$. Of course, it has long been known that the XXZ chain is integrable, but the result is more general, as it requires only the algebra (1, 2). Indeed, [34, 35] used this algebra to show that the "XXC models" are integrable. Similarly, the integrability of the Rydberg-blockade ladder we describe below is far from obvious without this method.

## 3.2 The XXZ spin chain

The XXZ chain (4) provides the best-known Hamiltonian of the form (3). Our approach to XXZ is not the usual Temperley-Lieb one: here $\Delta$ is allowed to vary freely even though the quantum-group parameter $q = e^{i\pi/3}$. The conventional formula $\Delta = q + q^{-1}$ does *not* apply here.

The XXZ Hamiltonian has $U(1)$ and $\mathbb{Z}_2$ symmetries, whose generators are

$$Q = \sum_{j=1}^{L} Z_{j+\frac{1}{2}} = n_\uparrow - n_\downarrow, \qquad F = \prod_{j=1}^{L} X_{j+\frac{1}{2}}, \tag{46}$$

where $n_\uparrow$ and $n_\downarrow$ are respectively the number of up and down spins in each configuration in the $Z$-diagonal basis. The eigenvalues of the charge $Q$ are therefore even integers for even $L$, and odd integers for odd $L$. These two symmetry generators anticommute with each other. In the Heisenberg chains at $\Delta = \pm 1$, the symmetries are enhanced to a full $SU(2)$.

## 3.3 The three-state antiferromagnet

In section 2.2, we showed how the algebra (1, 2) can be presented in terms of the chrome-plus algebra at $Q = 3$. The chromatic polynomial has long been known to be associated with the $Q$-state Potts model (see e.g. [47]), with the lines governed by the chromatic algebra [19] corresponding to domain walls in the Potts model. The degrees of freedom in the integer-$Q$ Potts model are "spins" taking $Q$ distinct values, such that a domain wall separates regions of different spins. In the chromatic interpretation, these values are the $Q$ different allowed colours. Here we show how to use this picture to find a three-state antiferromagnetic Potts Hamiltonian of the form (3).

In (13) we have a graphical presentation of the operators $P_j$ and $S_j$. Interpreting the lines as domain walls defines another presentation in terms of $Q = 3$ Potts spins, where these operators act on a Hilbert space comprised of $L$ three-state systems. The spins live on the dual graph to that made by the lines, so that we can define the operators via

$$P_j = \quad s_{j-1} \; \overset{s'_j}{\underset{s_j}{\times}} \; s_{j+1} \quad , \qquad\qquad S_j = \quad s_{j-1} \; \overset{s'_j}{\underset{s_j}{\rangle\!\langle}} \; s_{j+1} \quad . \tag{47}$$

A domain wall between two spins means that they they must be different, so we define basis states by specifying $s_j = A, B$ or $C$ for $j = 1 \ldots L$ such that $s_j \neq s_{j+1}$. Each operator $P_j$ or $S_j$ in general depends on three successive spins $s_{j-1}$, $s_j$ and $s_{j+1}$, and possibly changes the middle one to a value $s'_j$ while leaving the other two invariant. The operators therefore can be written in the form

$$O_j|s_{j-1}s_js_{j+1}\rangle = \sum_{s'_j} \quad s_{j-1} \; \overset{s'_j}{\underset{s_j}{\langle O \rangle}} \; s_{j+1} \quad |s_{j-1}s'_js_{j+1}\rangle, \tag{48}$$

where the matrix elements (pictured as a square) for $P_j$ and $S_j$ are determined from (47). The vertical line in $P_j$ requires $s_{j+1} \neq s_{j-1}$, while the horizontal line in $S_j$ requires $s_j \neq s'_j$. Thus

$$s_{j-1} \; \overset{s'_j}{\underset{s_j}{\langle P \rangle}} \; s_{j+1} = \big|\epsilon_{s_j s_{j-1} s_{j+1}}\big| \delta_{s_j s'_j}, \qquad s_{j-1} \; \overset{s'_j}{\underset{s_j}{\langle S \rangle}} \; s_{j+1} = \big|\epsilon_{s_j s_{j-1} s'_j}\big| \delta_{s_{j-1} s_{j+1}}, \tag{49}$$

where $\left|\epsilon_{abc}\right| = 1$ for $a, b, c \in \{A, B, C\}$ obeying $a \neq b \neq c \neq a$ and zero otherwise. The operator $P_j$ is thus diagonal in this basis, while $S_j$ is off-diagonal. Any pair of operators $O_j$ and $O'_{j'}$ satisfying the form (48) automatically commute for $|j - j'| > 1$, so (1) is automatically satisfied here and for all the presentations considered in this section. Checking that $P_j$ and $S_j$ defined by (49) do indeed satisfy the algebra (2) as well is straightforward to do.

The antiferromagnetic constraints $s_j \neq s_{j+1}$ can be displayed pictorially by an *adjacency diagram*. Each degree of freedom (i.e. each basis element at a site) corresponds to a node of the diagram, and nodes are connected by an edge if the corresponding spins are allowed to be adjacent to each other. Here the adjacency diagram is

$$
\begin{array}{c}
\text{(diagram of triangle with nodes } A, B, C\text{)}
\end{array}
\tag{50}
$$

The Hilbert space can be restricted to only states satisfying this constraint for all $j$, interpreted periodically in terms of spins. The dimension of this constrained Hilbert space $\mathcal{H}_3$ is then $2^L + 2(-1)^L$ with the $2^L$ arising from the two choices of $s_{j+1}$ given a particular $s_j$, and the correction from the periodic boundary conditions. The classical Potts model with the requirement that nearest-neighbour spins be different is called a zero-temperature antiferromagnet.

The next-nearest-neighbour Hamiltonian $H_3$ acting on $\mathcal{H}_3$ follows immediately from (3) by using the matrix elements in (49). To write it in operator form, we define $\mathbb{Z}_3$ analogs of the Pauli operators, $\sigma_j$ and $\tau_j$. They act non-trivially on the three-state system at site $j$, i.e. $\sigma_j = 1 \otimes 1 \cdots \otimes \sigma \otimes \dots 1$ and $\tau_j = 1 \otimes 1 \cdots \otimes \tau \otimes \dots 1$, where in the $ABC$ basis of $\mathcal{H}_3$

$$
\sigma = \begin{pmatrix} 1 & 0 & 0 \\ 0 & \omega & 0 \\ 0 & 0 & \omega^2 \end{pmatrix}, \qquad \tau = \begin{pmatrix} 0 & 0 & 1 \\ 1 & 0 & 0 \\ 0 & 1 & 0 \end{pmatrix}, \qquad \omega = e^{\frac{2\pi i}{3}}.
\tag{51}
$$

They satisfy $\sigma_j \tau_j = \omega \tau_j \sigma_j$ and $\sigma_j^3 = \tau_j^3 = 1$. Then

$$
H_3 = \frac{1}{9} \sum_{j=1}^{L} \Big( \tau_j \big(1 + \omega \sigma_{j-1}^\dagger \sigma_j + \omega^2 \sigma_{j-1} \sigma_j^\dagger \big) \big(1 + \omega^2 \sigma_j^\dagger \sigma_{j+1} + \omega \sigma_j \sigma_{j+1}^\dagger \big)
$$
$$
+ 3\Delta \big(1 - \sigma_{j-1} \sigma_{j+1}^\dagger \big) + \text{h.c.} \Big).
\tag{52}
$$

It is more illuminating to describe this Hamiltonian in words. The first part of the Hamiltonian corresponds to the projector $S_j$ of the algebra, and the second part with coefficient $3\Delta$ corresponds to the $P_j$ projector. As apparent from the matrix elements (49), $S_j$ toggles the spin $s_j$ whenever $s_{j-1} = s_{j+1}$, e.g. sends $ABA \leftrightarrow ACA$, whereas $P_j$ gives an energy whenever $s_{j-1} \neq s_{j+1}$.

This Hamiltonian has an obvious $S_3$ symmetry, corresponding to permutations of the three states in the $ABC$ basis, the symmetry of the adjacency diagram (50). The $\mathbb{Z}_3$ subgroup is generated by $R = \prod_j \tau_j$, which shifts all $s_j$ as $A \to B \to C \to A$. The other elements of $S_3$ exchange two of the states while leaving the third invariant. A less-obvious $U(1)$ symmetry becomes apparent with the mapping to XXZ: the difference of the two types of domain walls is conserved. Its generator is

$$
Q_3 = \frac{i}{\sqrt{3}} \sum_{j=1}^{L} \big( \sigma_j \sigma_{j+1}^\dagger - \sigma_j^\dagger \sigma_{j+1} \big),
\tag{53}
$$

commuting with $R$ but anticommuting with the generators of the $\mathbb{Z}_2$ subgroups of the $S_3$.

Antiferromagnetic Potts Hamiltonians like $H_3$ have not been widely studied. The special case $\Delta = -\frac{1}{2}$ does arise in the zero-temperature antiferromagnetic three-state classical Potts

model on the square lattice. This model was mapped onto the classical six-vertex model with identical Boltzmann weights for all configurations [21, 48–50]. This correspondence has been rederived in the categorical approach in [51], and follows in our setup by utilising the map $\mathcal{M}$ defined in section 4.2 on the classical three-state model. Taking the quantum Hamiltonian limit of the latter (see e.g. [45]) yields $H_{XXZ}$ with $\Delta = -\frac{1}{2}$. Varying $\Delta$ away from this value as we do corresponds to interactions between classical Potts spins at opposite corners of the same square in the classical model. While the zero-temperature constraint is required for the map $\mathcal{M}$ to make sense, relaxing the constraint in the quantum Hamiltonian version does not immediately change the physics [52], even though it does in the classical square-lattice model [21].

## 3.4 The integrable Rydberg-blockade ladder

Several other Hamiltonians with generators obeying (2) can be found by exploiting the connection to fusion categories. As explained in section 2.4, a fusion category enhances the graphical structure by relating it to a set of objects and their tensor products. Relating fusion categories to lattice models has a long history both in the study of integrable models (see e.g. [15] for a review) and the computation of topological invariants via "shadow world" [53–55]. Lattice models constructed in this fashion satisfy a number of remarkable properties, in particular allowing the construction of topological defects commuting with the Hamiltonian [6,7], results we exploit in section 4.

The allowed degrees of freedom in a lattice model built from a fusion category are labeled by the objects in the category. We call these degrees of freedom "heights" $h_j$, and they live on the sites $j = 1 \ldots L$ of our quantum chain, similarly to the Potts spins described above. A lattice model is then defined by specifying a particular object $\rho$ in the category. The objects labelling adjacent heights $h_{j+1}$ and $h_j$ then must satisfy the fusion rule $h_{j+1} \in h_j \otimes \rho$, i.e. $N_{h_j \rho}^{h_{j+1}} \neq 0$. For the $su(2)_k$ category with $\rho = \frac{1}{2}$, one obtains the RSOS models of Andrews, Baxter and Forrester [56]. In our cases, the $so(3)_2$ and $su(2)_4$ categories, we set $\rho = 1$, the spin-1 object, giving models introduced in [42]. The ensuing rules for heights are conveniently displayed in an adjacency diagram like (50). Using $\rho = 1$ and the $su(2)_4$ fusion rules from (34) gives the adjacency diagrams

$$
\begin{array}{ccc}
\bullet \!\!-\!\! \overset{\frown}{\bullet} \!\!-\!\! \bullet & \qquad & \overset{\frown}{\bullet} \!\!-\!\! \overset{\frown}{\bullet} \\
0 \quad 1 \quad 2 & & \tfrac{1}{2} \quad \tfrac{3}{2}
\end{array}
\tag{54}
$$

The adjacency diagram for $\rho = 1$ here splits into two disconnected pieces, as fusion with the spin-1 object does not mix integer and half-integer spins. The first diagram in (54) is that for the $so(3)_2$ subcategory of integer-spin objects, the fusion category subsuming the chrome-plus algebra.

Since the adjacency diagram splits into two disconnected parts, we can define two distinct height models. We first analyse the model with heights labelled by the integer spins 0, 1, 2 and adjacency rules displayed in the first diagram in (54) [22–25]. The fusion-category approach gives a general and explicit method for constructing the projection operators acting on the heights. The results are easiest to display in terms of the non-vanishing matrix elements. For $P_j^{(0)} = E_j/2$ they are

$$
0 \overset{1}{\underset{1}{\diamond(0)\diamond}} 0 \;=\; 2 \overset{1}{\underset{1}{\diamond(0)\diamond}} 2 \;=\; 1, \qquad 1 \overset{h_j'}{\underset{h_j}{\diamond(0)\diamond}} 1 \;=\; \tfrac{1}{4}\sqrt{d_{h_j} d_{h_j'}},
\tag{55}
$$

for $h_j, h'_j = 0, 1, 2$, while the quantum dimensions are $d_0 = d_2 = 1$ and $d_1 = 2$. The non-vanishing matrix elements for $P_j^{(2)}$ are

$$2 \,(2)\, 0 \;=\; 0 \,(2)\, 2 \;=\; 1\,, \qquad 1 \,(2)\, 1 \;=\; \tfrac{1}{4}(-1)^{h_j+h'_j}\sqrt{d_{h_j} d_{h'_j}}\,. \tag{56}$$

Using $S_j = P_j^{(0)} - P_j^{(2)}$ and $P_j = 1 - P_j^{(0)} - P_j^{(2)}$, it is straightforward to work out these matrix elements and to check that these operators satisfy (2). This presentation of $S_j$ and $P_j$ therefore gives an integrable Hamiltonian (3) for all $\Delta$.

A nice physical application of this chain arises in (experimentally realisable [26]) arrays of Rydberg-blockade atoms. Each site of such an array is a two-state system, with the states labelled empty and occupied. The interactions can be tuned to forbid nearby sites from both being occupied, hence the "blockade". The case of a chain with forbidden nearest-neighbour occupancy [57] has been studied intensively recently, both because of the experimental interest as well providing an example of "quantum scars" [27]. The chain studied here can be interpreted as the *integrable Rydberg-blockade ladder* (IRL). Namely, consider a ladder with $L$ rungs indexed by $j$. We identify the three states for each $j$ as having a particle occupying a site on the top of the rung (state 0), one occupying the bottom of the rung (state 2), and an empty rung (state 1). The adjacency diagram in (54) thus forbids particles on adjacent rungs, no matter whether the top or bottom of the rung is occupied. Thus the IRL allows for just one particle per square on the ladder. The dimension of the corresponding Hilbert space $\mathcal{H}_{\text{IRL}}$ is $2^L + (-1)^L$.

We can write the Hamiltonian in explicit operator form by defining the operators $t_j^\dagger$ and $b_j^\dagger$ that create a boson on the top and bottom of rung $j$ respectively, subject to rungs $j+1$ and $j-1$ being empty. Thus by definition, $t_j = n_{j-1}^{(e)} t_j n_{j+1}^{(e)}$ and likewise for $b_j$. In the height language, they take $|111\rangle \to |101\rangle$ and $|111\rangle \to |121\rangle$ respectively. The occupation numbers are $n_j^{(t)} = t_j^\dagger t_j$ and $n_j^{(b)} = b_j^\dagger b_j$ and $n_j^{(e)} = 1 - n_j^{(b)} - n_j^{(t)}$ for the three possibilities on rung $j$. Then

$$S_j = \tfrac{1}{\sqrt{2}}\big(t_j^\dagger + t_j + b_j^\dagger + b_j\big) + \big(n_{j-1}^{(t)} - n_{j-1}^{(b)}\big)\big(n_{j+1}^{(t)} - n_{j+1}^{(b)}\big)\,,$$
$$P_j = \tfrac{1}{2}\big(t_j^\dagger - b_j^\dagger\big)\big(t_j - b_j\big) + \big(n_{j-1}^{(e)} - n_{j+1}^{(e)}\big)^2\,. \tag{57}$$

The Hamiltonian is then found as always from (3).

It is illuminating to write $H_{\text{IRL}}$ in terms of different operators. Because the Rydberg exclusion here is one particle per square, it makes sense to utilise the operators $p_j = (t_j + b_j)/\sqrt{2}$ and $m_j = (t_j - b_j)/\sqrt{2}$. Thus instead of particles on the top and bottom of each rung, one can consider $+$ and $-$ states annihilated by $p$ and $m$ respectively, both still forbidding particles on neighbouring rungs. The number operators and the swap operator are

$$n_j^+ = p_j^\dagger p_j\,, \qquad n_j^- = m_j^\dagger m_j\,, \qquad s_j = p_j^\dagger m_j + m_j^\dagger p_j\,. \tag{58}$$

One convenient feature of the $\pm$ basis is that the operator $P_j$ is diagonal, giving the Hamiltonian

$$H_{\text{IRL}} = \sum_j \left[ p_j + p_j^\dagger + s_{j-1}s_{j+1} + \Delta\Big(n_j^- + \big(n_{j-1}^{(e)} - n_{j+1}^{(e)}\big)^2\Big)\right]. \tag{59}$$

Another convenience is that only the $+$ particles can be created individually; the only way of creating/annihilating $-$ particles is as a pair two rungs apart. The Hamiltonian therefore has two $\mathbb{Z}_2$ symmetries when $L$ is even, generated by

$$\widetilde{F}_{\text{odd}} = (-1)^{\sum_{j\,\text{odd}} n_j^-}\,, \qquad \widetilde{F}_{\text{even}} = (-1)^{\sum_{j\,\text{even}} n_j^-}\,. \tag{60}$$

Although it appears a bit daunting at first glance, we describe in sections 5.3 and 6 how the physics of the IRL follows from that of XXZ, in particular being critical for $|\Delta| \leq 1$. We discuss the other phases below and what happens under perturbation in our companion paper [30].

### 3.5 Ising zigzag ladder

The fourth Hamiltonian studied in this paper comes from the second adjacency diagram drawn in (54). Each height is either $\frac{1}{2}$ or $\frac{3}{2}$, so that each integer-labelled site $j$ is simply a two-state system, with no further constraints from the adjacency rules: either state can be adjacent to itself or the other. The Hilbert space $\mathcal{H}_{\text{zig}}$ thus is $2^L$-dimensional. The matrix elements can be found by utilising the results of section 7.4 of [7] (where this model is called $\mathcal{M}_1$). They are

$$
h_{j-1} \;\substack{h'_j \\ (0) \\ h_j}\; h_{j+1} = \delta_{h_{j-1}h_{j+1}}, \qquad h_{j-1} \;\substack{h'_j \\ (2) \\ h_j}\; h_{j+1} = \delta_{h_{j-1},\,2-h_{j+1}}. \tag{61}
$$

The projectors can be written in operator form utilising the Pauli matrices, which to avoid confusion with XXZ we label as $\sigma_j^a$ with $a = x, y, z$. Then

$$
P_j^{(0)} = \tfrac{1}{4}(1 + \sigma_{j-1}^z \sigma_{j+1}^z)(1 + \sigma_j^x), \qquad P_j^{(2)} = \tfrac{1}{4}(1 - \sigma_{j-1}^z \sigma_{j+1}^z)(1 + \sigma_j^x). \tag{62}
$$

It is easy to verify that the ensuing $S_j = P_j^{(0)} - P_j^{(2)}$ and $P_j = 1 - P_j^{(0)} - P_j^{(2)}$ satisfy (1, 2).

The Hamiltonian constructed from (3) using (62),

$$
H_{\text{zig}} = \tfrac{1}{2} \sum_{j=1}^{L} \left( \sigma_{j-1}^z \left(1 + \sigma_j^x\right) \sigma_{j+1}^z + \Delta(1 - \sigma_j^x) \right), \tag{63}
$$

is therefore integrable for any $\Delta$. As usual, we take periodic boundary conditions in (63), and so interpret indices cyclically. One can think of this Hamiltonian as describing two Ising chains, one on the even sites and the other on the odd, coupled via the $zxz$ terms. The chains are thus naturally written as a zigzag ladder. If one omits either the $zz$ or the $zxz$ terms the model is free fermion, and we have shown that when their coefficients are equal the model is integrable if not free. It is worth noting that the $zxz$ terms arise as a canonical example of a symmetry-protected topological phase [58], and $H_{\text{zig}}$ has been studied in a variety of interesting contexts [59–61]. As with the Rydberg ladder, $H_{\text{zig}}$ has a $\mathbb{Z}_2 \times \mathbb{Z}_2$ symmetry for even $L$ generated by $F_{\text{odd}} = \prod_{j \text{ odd}} \sigma_j^x$ and $F_{\text{even}} = \prod_{j \text{ even}} \sigma_j^x$. An additional symmetry generator is $E$, the operator that interchanges the two chains. As $EF_1 = F_2E$, the symmetry group is the dihedral group $D_4$, the symmetry of a square. For odd $L$ only the product $F_{\text{zig}} = F_{\text{odd}}F_{\text{even}}$ commutes.

## 4 A square of non-invertible mappings

Because all Hamiltonians in the quartet are built from the same algebra, their physics must be related. In this section we make this notion precise by constructing explicit linear maps between them. In general, this requires some care, as the maps are non-invertible. We sum-

Table 1: The symmetries, the non-invertible maps, and their interrelations.

| Hamiltonian | Symmetry generators | Products of non-invertible maps |
|---|---|---|
| $H_{\text{XXZ}}$ | $Q$ ($U(1)$ charge); $F$ (spin flip) | $\mathcal{K}^\dagger \mathcal{K} = 1 + F$; $\mathcal{M}\mathcal{M}^\dagger = 1 + \omega^Q + \omega^{-Q}$ |
| $H_3$ | $Q_3$ ($U(1)$ charge); $R$, $F_3$ (permute spins) | $\mathcal{O}^\dagger \mathcal{O} = 1 + F_3$; $\mathcal{M}^\dagger \mathcal{M} = 1 + R + R^2$ |
| $H_{\text{zig}}$ | $\mathcal{Q}_{\text{zig}} = \mathcal{K}Q^2\mathcal{K}^\dagger$; $F_{\text{zig}} = (-1)^L F_{\text{odd}} F_{\text{even}}$ | $\mathcal{K}\mathcal{K}^\dagger = 1 + F_{\text{zig}}$; $\mathcal{D}\mathcal{D}^\dagger = 1 + (1 + F_{\text{zig}})\cos\frac{2\pi}{3}\sqrt{\mathcal{Q}_{\text{zig}}}$ |
| $H_{\text{IRL}}$ | $\widetilde{\mathcal{Q}}$; $\widetilde{\mathcal{S}} = \mathcal{O}R\mathcal{O}^\dagger$; $\widetilde{F} = (-1)^L \widetilde{F}_{\text{odd}}\widetilde{F}_{\text{even}}$ | $\mathcal{O}\mathcal{O}^\dagger = 1 + \widetilde{F}$; $\mathcal{D}^\dagger\mathcal{D} = 1 + \widetilde{\mathcal{S}}$ |

marise these non-invertible mappings in the diagram

$$
\begin{array}{ccc}
\mathcal{H}_3 & \xrightarrow{\mathcal{M}} & \mathcal{H}_{\text{XXZ}} \\
\mathcal{O} \downarrow & & \downarrow \mathcal{K} \\
\widetilde{\mathcal{Q}}, \widetilde{\mathcal{S}} \curvearrowright \mathcal{H}_{\text{IRL}} & \xrightarrow{\mathcal{D}} & \mathcal{H}_{\text{zig}} \curvearrowright \mathcal{Q}_{\text{zig}}
\end{array}
\qquad
\begin{array}{l}
\mathcal{H}_3\text{: 3-state antiferromagnet,} \\
\mathcal{H}_{\text{IRL}}\text{: integrable Rydberg-blockade ladder,} \\
\mathcal{H}_{\text{XXZ}}\text{: XXZ chain,} \\
\mathcal{H}_{\text{zig}}\text{: Ising zigzag ladder.}
\end{array}
\tag{64}
$$

In this section we define all the maps between models by utilising topological defects, making more explicit and extending the analyses of [7, 20]. We derive a variety of interrelations among these maps and conventional symmetries, in particular showing that the diagram of maps between models is commutative, i.e. $\mathcal{K}\mathcal{M} = \mathcal{D}\mathcal{O}$. The maps that send $\mathcal{H}_{\text{IRL}}$ and $\mathcal{H}_{\text{zig}}$ to themselves commute with the corresponding Hamiltonians, and so generate non-invertible symmetries that we analyse in section 5. We collect all our results for the interrelations between the symmetries and maps in Table 1.

## 4.1 XXZ and the Ising zigzag ladder

The non-invertible maps relating our four models are best displayed by using the formalism of topological defects. These maps are examples of *defect-creation operators* [7]. We start with the defect that maps between the XXZ chain and the Ising zigzag ladder, as it in essence implements the Kramers-Wannier duality [1] of the Ising lattice model. We find a map $\mathcal{K}: \mathcal{H}_{\text{XXZ}} \to \mathcal{H}_{\text{zig}}$ that commutes with the Hamiltonians in the sense that

$$
\mathcal{K}H_{\text{XXZ}} = H_{\text{zig}}\mathcal{K}, \qquad H_{\text{XXZ}}\mathcal{K}^\dagger = \mathcal{K}^\dagger H_{\text{zig}}. \tag{65}
$$

The Ising model has two non-trivial defect-creation operators. One is simply the spin-flip operator $F$. The duality-defect creation operator $\mathcal{D}_\sigma$ is much less obvious. In essence, it maps from domain walls to spins. Its matrix elements can be written as a product of local defect weights, each depending on $s_{j+\frac{1}{2}} = \uparrow, \downarrow$ in XXZ and one of the states $t_{j\pm 1} = \frac{1}{2}, \frac{3}{2}$ of zigzag Ising, giving [7]

$$
\left(\mathcal{D}_\sigma\right)_{\{s\}}^{\{t\}} = \prod_{j=1}^L w\left(s_{j-\frac{1}{2}}, t_j\right) w\left(s_{j+\frac{1}{2}}, t_j\right), \qquad w(s,t) = 2^{-\frac{1}{4}}(-1)^{\delta_{s,\downarrow}\delta_{t,\frac{3}{2}}}. \tag{66}
$$

Using (66) for a pair of adjacent spins in XXZ gives the height in the Ising zigzag ladder in between:

$$
\mathcal{D}_\sigma: \quad \{|\uparrow\uparrow\rangle, |\downarrow\downarrow\rangle\} \to \tfrac{1}{\sqrt{2}}\left(|\tfrac{1}{2}\rangle + |\tfrac{3}{2}\rangle\right) \equiv |+\rangle, \quad \{|\uparrow\downarrow\rangle, |\downarrow\uparrow\rangle\} \to \tfrac{1}{\sqrt{2}}\left(|\tfrac{1}{2}\rangle - |\tfrac{3}{2}\rangle\right) \equiv |-\rangle, \tag{67}
$$

where $|\pm\rangle$ have eigenvalues $\pm 1$ under $\sigma_j^x$. Similarly,

$$
\mathcal{D}_\sigma^\dagger: \quad \{|\tfrac{1}{2}\tfrac{1}{2}\rangle, |\tfrac{3}{2}\tfrac{3}{2}\rangle\} \to \tfrac{1}{\sqrt{2}}\left(|\uparrow\rangle + |\downarrow\rangle\right), \qquad \{|\tfrac{1}{2}\tfrac{3}{2}\rangle, |\tfrac{3}{2}\tfrac{1}{2}\rangle\} \to \tfrac{1}{\sqrt{2}}\left(|\uparrow\rangle - |\downarrow\rangle\right), \tag{68}
$$

Thus under Kramers-Wannier duality, a domain wall in the $z$-basis of one model becomes a down spin in the $x$-basis of the other, while no domain wall becomes an up spin. The defect-creation operators therefore satisfy

$$\mathcal{D}_\sigma Z_{j-\frac{1}{2}} Z_{j+\frac{1}{2}} = \sigma_j^x \mathcal{D}_\sigma, \qquad \mathcal{D}_\sigma X_{j+\frac{1}{2}} = \sigma_j^z \sigma_{j+1}^z \mathcal{D}_\sigma. \tag{69}$$

Acting with $\mathcal{D}_\sigma$ therefore implements Kramers-Wannier duality in the Ising Hamiltonian by effectively exchanging each transverse-field and interaction term on the integer sites for the interaction and transverse-field terms respectively on the half-integer sites. These relations are an example of the defect commutation relations described below in (94).

Since the terms in the Hamiltonians of the XXZ chain (4) and the Ising zigzag ladder (63) can be built from products of the terms in (69) the maps can be used here as well. Defining

$$\mathcal{K} = \left( \prod_{j=1}^L \sigma_j^z \right) \mathcal{D}_\sigma, \tag{70}$$

the analogs of (67) and (69) are

$$\mathcal{K}: \quad \{|\uparrow\uparrow\rangle, |\downarrow\downarrow\rangle\} \to |-\rangle, \qquad \{|\uparrow\downarrow\rangle, |\downarrow\uparrow\rangle\} \to |+\rangle, \tag{71}$$

$$\mathcal{K} Z_{j-\frac{1}{2}} Z_{j+\frac{1}{2}} = -\sigma_j^x \mathcal{K}, \qquad \mathcal{K} X_{j-\frac{1}{2}} X_{j+\frac{1}{2}} = \sigma_{j-1}^z \sigma_{j+1}^z \mathcal{K}. \tag{72}$$

Using (72) with the Hamiltonians (4) and (63) yields (65).

As with all the non-invertible maps we study, $\mathcal{K}$ and $\mathcal{K}^\dagger$ are non-trivial only in certain subsectors. Indeed, since periodic boundary conditions require an even number of domain walls in XXZ, the image of $\mathcal{K}$ from (70) must have an even number of + spins. This sector is invariant under spin-flip symmetry operator $F_{\text{zig}} \equiv (-1)^L \prod_j \sigma_j^x$. Indeed, the definitions (66) and (70) require $F_{\text{zig}} \mathcal{K} = \mathcal{K} = \mathcal{K} F$, so that $\mathcal{K}$ acts non-trivially only on the XXZ sector invariant under $F$.

A more systematic method of determining the images and the kernels of the maps comes from the *fusion rules* of the defect-creation operators. In the fusion-category construction, these automatically follow from the fusion rules of the objects. In this Ising case they are [5]

$$\mathcal{D}_\sigma \mathcal{D}_\sigma^\dagger = 1 + (-1)^L F_{\text{zig}}, \qquad \mathcal{D}_\sigma^\dagger \mathcal{D}_\sigma = 1 + F. \tag{73}$$

Both rules are easily checked explicitly. The modified map (70) therefore satisfies

$$\boxed{\mathcal{K} \mathcal{K}^\dagger = 1 + F_{\text{zig}}, \qquad \mathcal{K}^\dagger \mathcal{K} = 1 + F.} \tag{74}$$

The maps are thus indeed non-trivial in the sectors with eigenvalue 1 of $F_{\text{zig}}$ and of $F$. The corresponding partition functions therefore are related as

$$\text{tr}\left[ e^{-\beta H_{\text{zig}}} (1 + F_{\text{zig}}) \right] = \text{tr}\left[ e^{-\beta H_{\text{XXZ}}} (1 + F) \right]. \tag{75}$$

One advantage of the defect approach is that it is easy to find maps between other sectors. For example, the fusion rules can be modified by removing $\sigma_L^z$ in the products in (70):

$$\mathcal{K}_- = \sigma_L^z \mathcal{K} \quad \Longrightarrow \quad \mathcal{K}_- \mathcal{K}_-^\dagger = 1 - F_{\text{zig}}, \qquad \mathcal{K}_-^\dagger \mathcal{K}_- = 1 + F. \tag{76}$$

The modified maps acting to and from $\mathcal{H}_{\text{zig}}$ are non-vanishing only in the odd sector of $F_{\text{zig}}$. Between the two pairs of maps, we have covered all of $\mathcal{H}_{\text{zig}}$. However, the commutation relations analogous to (72) are

$$\mathcal{K}_- Z_{j-\frac{1}{2}} Z_{j+\frac{1}{2}} = -(-1)^{\delta_{jL}} \sigma_j^x \mathcal{K}_-, \qquad \mathcal{K}_- X_{j-\frac{1}{2}} X_{j+\frac{1}{2}} = \sigma_{j-1}^z \sigma_{j+1}^z \mathcal{K}_-. \tag{77}$$

Because of the factor $(-1)^{\delta_{jL}}$ here, the modified maps no longer involve $H_{\text{XXZ}}$. Instead,

$$\mathcal{K}_- H_- = H_{\text{zig}} \mathcal{K}_-, \qquad H_- \mathcal{K}_-^\dagger = \mathcal{K}_-^\dagger H_{\text{zig}}, \tag{78}$$

where the XXZ Hamiltonian with a "$\mathbb{Z}_2$-twisted" boundary condition is

$$H_- = \tfrac{1}{2} \sum_{j=1}^{L} \left( X_{j-\frac{1}{2}} X_{j+\frac{1}{2}} + (-1)^{\delta_{jL}} Y_{j-\frac{1}{2}} Y_{j+\frac{1}{2}} + \Delta \left( 1 + (-1)^{\delta_{jL}} Z_{j-\frac{1}{2}} Z_{j+\frac{1}{2}} \right) \right). \tag{79}$$

The analogous boundary condition in Ising following from (77) is called "anti-periodic" [62].

The map $\mathcal{K}_-$ relates partition functions as

$$\text{tr}\left[ e^{-\beta H_{\text{zig}}} (1 - F_{\text{zig}}) \right] = \text{tr}\left[ e^{-\beta H_-} (1 + F) \right]. \tag{80}$$

Combining this relation with (75) allows the partition function of our zigzag Ising chain to be written as a sum over partition functions in XXZ:

$$Z_{\text{zig}} = \text{tr}\left[ e^{-\beta H_{\text{zig}}} \right] = \tfrac{1}{2} \text{tr}\left[ \left( e^{-\beta H_{\text{XXZ}}} + e^{-\beta H_-} \right) (1 + F) \right]. \tag{81}$$

One can derive similar linear relations between the XXZ and twisted zigzag Ising partition functions. We emphasise that such identities are exact. If a model has a continuum limit, then they necessarily hold there as well, as for the conformal field theories described in section 6.

## 4.2 XXZ and the three-state antiferromagnet

Here we construct the map $\mathcal{M} : \mathcal{H}_3 \to \mathcal{H}_{\text{XXZ}}$ that satisfies the "commutation" relation

$$\mathcal{M} H_3 = H_{\text{XXZ}} \mathcal{M}, \qquad H_3 \mathcal{M}^\dagger = \mathcal{M}^\dagger H_{\text{XXZ}}. \tag{82}$$

This construction generalises the correspondence of [21] to any $\Delta$.

As with Kramers-Wannier duality, $\mathcal{M}$ maps domain walls in the three-state antiferromagnetic chain to spins in the XXZ chain. Because of the restriction that adjacent spins be different in the $ABC$ basis of $\mathcal{H}_3$, there are six different allowed nearest-neighbour pairs $(s_j, s_{j+1})$. We define two types of domain walls by identifying configurations related by the $\mathbb{Z}_3$ symmetry $R$. Each type is mapped to one of the two states in the $Z$-diagonal basis in XXZ, namely

$$\mathcal{M} : \quad \left\{ |AB\rangle, |BC\rangle, |CA\rangle \right\} \to |\uparrow\rangle, \qquad \left\{ |AC\rangle, |BA\rangle, |CB\rangle \right\} \to |\downarrow\rangle. \tag{83}$$

The map $\mathcal{M}$ is then defined by applying (83) to send each domain wall $(s_j, s_{j+1})$, including the "round the world" pair with $j = L$, to obtain the XXZ spin at site $j + \frac{1}{2}$ (mod $L$). It is then easy to check that the "commutation" relation (82) is satisfied with $H_3$ defined by (52).

The map $\mathcal{M}$ is $3 \to 1$ in these bases, as the three configurations related by $R$ all map to the same XXZ configuration, i.e.

$$\mathcal{M} R = \mathcal{M} \quad \Longrightarrow \quad \tfrac{1}{3} \mathcal{M} (1 + R + R^2) = \mathcal{M}. \tag{84}$$

The latter means $\mathcal{M}$ is non-vanishing only on the sector invariant under $R$ and so acts nontrivially on only a third of $\mathcal{H}_3$ (its dimension $2^L + 2(-1)^L$ is always divisible by three). Since the dimension of $\mathcal{H}_{\text{XXZ}}$ is $2^L$, not all states in $\mathcal{H}_{\text{XXZ}}$ are in the image of $\mathcal{M}$.

The $U(1)$ symmetries defined by (46) and (53) enable us to understand the image of $\mathcal{M}$. The explicit form of (83) requires the two are related by $\mathcal{M} Q_3 = Q \mathcal{M}$. The charge $Q_3$ is diagonal in the $ABC$ basis, and the periodic boundary conditions require the eigenvalues to be $0 \bmod 3$. The eigenvalues of $Q$ in the image of $\mathcal{M}$ must obey the same constraint, so

$$\boxed{\mathcal{M}^\dagger \mathcal{M} = 1 + R + R^2, \qquad \mathcal{M} \mathcal{M}^\dagger = 1 + \omega^Q + \omega^{-Q},} \tag{85}$$

as is straightforward to check explicitly using (83). The partition functions of the two models are then related by projecting onto the appropriate sectors. Defining

$$Z_3[a] \equiv \text{tr}\left[e^{-\beta H_3}R^a\right], \qquad Z_{\text{XXZ}}(\nu) \equiv \text{tr}\left[e^{-\beta H_{\text{XXZ}}}\omega^{\nu Q}\right], \tag{86}$$

and utilising (85) gives

$$Z_3[0] + Z_3[1] + Z_3[2] = Z_{\text{XXZ}}(0) + Z_{\text{XXZ}}(1) + Z_{\text{XXZ}}(2). \tag{87}$$

Extending the maps non-trivially to other sectors requires defining $\mathbb{Z}_3$-twisted boundary conditions for XXZ. The modified map is defined analogously to (76) as

$$\mathcal{M}_\omega = \mathcal{M}\sigma_L \quad \Longrightarrow \quad \mathcal{M}_\omega^\dagger \mathcal{M}_\omega = 1 + \omega^2 R + \omega R^2, \qquad \mathcal{M}_\omega \mathcal{M}_\omega^\dagger = 1 + \omega^Q + \omega^{-Q}, \tag{88}$$

where $\sigma_L$ is defined in (51). Thus $\mathcal{M}_\omega$ acts non-trivially only on the sector where $R$ has eigenvalue $\omega$. It "commutes" with the Hamiltonians as $\mathcal{M}_\omega H_3 = H_\omega \mathcal{M}_\omega$, where the XXZ Hamiltonian with $\mathbb{Z}_3$-twisted boundary conditions is

$$H_\omega = \sum_{j=1}^{L}\left(\omega^{-\delta_{jL}}S^+_{j-\frac{1}{2}}S^-_{j+\frac{1}{2}} + \omega^{\delta_{jL}}S^-_{j-\frac{1}{2}}S^+_{j+\frac{1}{2}} + \tfrac{1}{2}\Delta\left(1 + Z_{j-\frac{1}{2}}Z_{j+\frac{1}{2}}\right)\right), \tag{89}$$

with $S^\pm_j = (X_j \pm iY_j)/2$ the usual raising and lowering operators. The corresponding partition functions are then related as

$$Z_3[0] + \omega^2 Z_3[1] + \omega Z_3[2] = Z_\omega(0) + Z_\omega(1) + Z_\omega(2), \quad \text{where } Z_\omega(\nu) = \text{tr}\left[e^{i\frac{2}{3}\pi\nu}e^{-\beta H_\omega}\right]. \tag{90}$$

The map $\mathcal{M}_{\omega^2} = \mathcal{M}\sigma_L^\dagger$ goes from the sector with $R = \omega^2$ to the conjugate $\mathbb{Z}_3$-twisted XXZ Hamiltonian $H_{\omega^2}$, with the corresponding partition function identity given by (90) with $\omega \to \omega^2$. We then have, analogously to (81), the identities

$$Z_3[a] = \tfrac{1}{3}\sum_{\nu=0,1,2}\left(Z_{\text{XXZ}}(\nu) + \omega^{-a\nu}Z_\omega(\nu) + \omega^{a\nu}Z_{\omega^2}(\nu)\right). \tag{91}$$

One can also obtain sectors with different $Q$ charges by modifying the Hilbert space $\mathcal{H}_3$ to allow identical neigbours across one link, with a corresponding modification of the Hamiltonian $H_3$.

## 4.3 The Rydberg ladder and the three-state antiferromagnet

Constructing the maps to and from the integrable Rydberg-blockade ladder (IRL) displayed in (64) requires a little more technology. The matrix elements here are written in terms of local defect weights that depend on two adjacent degrees of freedom in each of the models:

$$W^{h_j,h_{j+1}}_{s_j,s_{j+1}} = \begin{array}{c} h_j \qquad h_{j+1} \\ \boxed{\phantom{xx}} \\ s_j \qquad s_{j+1} \end{array}, \tag{92}$$

with the maps then given by the product

$$\mathcal{O}^{\{h\}}_{\{s\}} = \prod_{j=1}^{L} W^{h_j,h_{j+1}}_{s_j,s_{j+1}} = \begin{array}{c} h_1 \quad h_2 \quad h_3 \qquad\qquad h_L \\ \cdots \\ s_1 \quad s_2 \quad s_3 \qquad\qquad s_L \end{array}. \tag{93}$$

We have displayed here the map $\mathcal{O}: \mathcal{H}_3 \to \mathcal{H}_{\text{IRL}}$ with degrees of freedom those of the Rydberg ladder (heights $h_j = 0,1,2$) and the three-state antiferromagnet (spins $s_j = A, B, C$),

but all the non-invertible maps we discuss can be written in this form. For example, for the Kramers-Wannier duality map $\mathcal{D}_\sigma$, this form is utilised in [5], albeit with only two degrees of freedom per rectangle. To guarantee that the "commutation" relations such as (82) and (65) are satisfied, it is sufficient to require that the matrix elements satisfy the *defect commutation relations* [7]. They are local conditions on the defect weights written in pictorial form as

$$
\sum_{s'_j}
\begin{array}{c}
h_j \\
h_{j-1} \quad h_{j+1} \\
s_{j-1} \quad s_{j+1} \\
s_j
\end{array}
=
\sum_{h'_j}
\begin{array}{c}
h_j \\
h_{j-1} \quad h_{j+1} \\
s_{j-1} \quad s_{j+1} \\
s_j
\end{array}
,
\tag{94}
$$

where the square represents the matrix elements of any of the projectors $P^{(a)}$ in the respective models. The sums over the internal degrees of freedom $s'_j$ and $h'_j$ on the two sides is essential; the equality is not in general between individual matrix elements. Dragging this defect through toggles between the Hilbert spaces on which the projectors act, as for example in (69).

We find the map $\mathcal{O}$ using a lattice version [33] of the orbifold construction of conformal field theory [31]. The lattice orbifold requires no criticality, but like its continuum analog, in essence gauges a discrete symmetry. Any degrees of freedom related by the symmetry are identified, but new degrees of freedom arise as well. In the lattice version, one identifies distinct states related by the symmetry, but then any states invariant under the symmetry turn into a multiplet [33, 63, 64].

We orbifold the three-state antiferromagnet using the $\mathbb{Z}_2$ symmetry $F_3$ exchanging state $B \leftrightarrow C$ while leaving $A$ invariant. The case at hand has already been described in [33], under the name of $\widehat{A}_5 \leftrightarrow \widehat{D}_5$. (The names come from identifying the adjacency diagrams as extended Dynkin diagrams; one obtains those in [33] from ours in (50, 54) by giving different labels to the heights on even and odd sites.) In the adjacency diagram (50), $F_3$ corresponds to a vertical reflection.

The first step is to identify the adjacency diagram of the orbifold model. Under this orbifold, $B$ and $C$ are identified into a height we label (with foresight) by 1, while $A$ doubles into two states, say called 0 and 2. The adjacency diagram is no longer a triangle after the orbifold. The edges $A$ shared with $B$ and with $C$ result in the orbifold in edges between 1 and 2 and between 1 and 0. The edge connecting $B$ and $C$ turns into an edge connecting 1 to itself. The orbifold adjacency diagram thus is precisely the Rydberg-ladder adjacency diagram, the first of (54). The procedure works in reverse: we could have just as well orbifolded the IRL by its $\mathbb{Z}_2$ symmetry $\widetilde{F}$, which exchanges 0 and 2 and so reflects its adjacency diagram horizontally. Then 0 and 2 are identified into $A$, while 1 is doubled into $B$ and $C$.

The orbifold construction immediately says which of the weights (92) are non-vanishing: for any $j$, if $s_j = A$, then $h_j \in 0, 2$, while if $s_j = B$ or $C$, then $h_j = 1$. To construct their precise values, there exists a generalisation of the construction of [7] to the orbifold case, using module categories [51]. However, here it is much simpler to reverse-engineer the defect weights directly from the orbifold Boltzmann weights derived in [33]. The case here is only marginally more complicated than Kramers-Wannier duality, with the non-vanishing defect weights

$$
\begin{array}{c}
1 \qquad 1 \\
\\
B \qquad C
\end{array}
= 1 \,,
\qquad\qquad
\begin{array}{c}
h \qquad 1 \\
\\
A \qquad s
\end{array}
= 2^{-\frac{1}{4}}(-1)^{\delta_{sC}\delta_{h2}} \,,
\tag{95}
$$

where $h = 0$ or 2 and $s = B$ or $C$, and all are invariant under both horizontal and vertical reflections. If one were to omit the edge connecting $B$ and $C$ and thus that connecting the height 1

to itself, the first weight here is zero, and the weights reduce to those given for $\mathcal{D}_\sigma$ in (66). The $\mathbb{Z}_2$ lattice orbifold is therefore a very natural generalisation of Kramers-Wannier duality [33].

We gain intuition into the ensuing non-invertible maps $\mathcal{O}$ and $\mathcal{O}^\dagger$ by combining the $0, 2$ states of the IRL into the combinations $|\pm\rangle = (|0\rangle \pm |2\rangle)/\sqrt{2}$ described in (58). This change of basis is possible because the one-particle-per-square rule requires that states 0 and 2 (a particle on the top and bottom of a rung respectively) must always be adjacent to 1 (an empty rung). For clarity here and later on, we label the state 1 by $e$ in this basis. Similarly to Kramers-Wannier duality, the image of a configuration in the $ABC$ basis of the Hilbert space $\mathcal{H}_3$ under $\mathcal{O}^\dagger$ is a unique state in the $\pm$ basis of $\mathcal{H}_{\text{IRL}}$. Namely, the defect weights (95) require that spins $B$ and $C$ always map to $e$, while $A$ maps to $\pm$ depending on the adjacent spins. Thus knowing three successive spins $(s_{j-1}, s_j, s_{j+1})$ in the antiferromagnet gives $h_j$ in the Rydberg ladder via

$$\mathcal{O}: \quad \{|B\rangle, |C\rangle\} \to |e\rangle, \quad \{|BAB\rangle, |CAC\rangle\} \to |+\rangle, \quad \{|BAC\rangle, |CAB\rangle\} \to |-\rangle. \tag{96}$$

The orbifold construction guarantees that the Hamiltonians are related as with the other maps. Indeed, using the $\pm$ basis here makes it straightforward to verify that both the maps $\mathcal{O}$ and $\mathcal{O}^\dagger$ satisfy the defect commutation relations (94), and thus satisfy

$$\mathcal{O}H_3 = H_{\text{IRL}}\mathcal{O}, \qquad H_3\mathcal{O}^\dagger = \mathcal{O}^\dagger H_{\text{IRL}}. \tag{97}$$

The definition (96) means that $\mathcal{O}F_3 = \mathcal{O}$, so that this map is non-trivial only on the sector invariant under $F_3$. Moreover, imposing periodic boundary conditions requires that its image lie in the sector where the eigenvalue of $\widetilde{F} \equiv (-1)^L \widetilde{F}_{\text{odd}} \widetilde{F}_{\text{even}} = (-1)^{\sum_j (1-n_j^-)}$ is 1. Then, as is easy to check explicitly, the fusion rules are

$$\boxed{\mathcal{O}\mathcal{O}^\dagger = 1 + \widetilde{F}, \qquad \mathcal{O}^\dagger\mathcal{O} = 1 + F_3.} \tag{98}$$

The maps are thus non-invertible as advertised, and the partition functions are related as

$$\text{tr}\left[e^{-\beta H_{\text{IRL}}}\left(1 + \widetilde{F}\right)\right] = \text{tr}\left[e^{-\beta H_3}\left(1 + F_3\right)\right]. \tag{99}$$

To obtain an expression for the Rydberg partition function analogous to (81), we define the $F_3$-twisted boundary condition in the antiferromagnet. With this boundary condition, we no longer set $s_{L+j} = s_j$, but instead $s_{L+j} = \bar{s}_j$, where $\bar{A} = A$, $\bar{B} = C$, and $\bar{C} = B$. The Hilbert space becomes $\mathcal{H}_{3-}$, where $s_L \neq s_1$ is not required, but rather $\bar{s}_L \neq s_1$. The corresponding Hamiltonian $H_{3-}$ is then built from (49) as before, but using the appropriate bars in the matrix elements with $j = L$ or $j = 1$. Namely, in $S_1$ and $P_1$ one uses the spins $(h_0, h_1, h_2) = (\bar{h}_L, h_1, h_2)$, while in $S_L$ and $P_L$ one uses $(h_{L-1}, h_L, \bar{h}_1)$. The map $\mathcal{O}_- : \mathcal{H}_{3-} \to \mathcal{H}_{\text{IRL}}$ is built using (96) as before, with the same barred spins when the map wraps around. For example, $\mathcal{O}_-|ABC\rangle = |+ee\rangle$, $\mathcal{O}_-|BAB\rangle = |e+e\rangle$ and $\mathcal{O}_-|CABC\rangle = |e-ee\rangle$. As these examples illustrate, $\mathcal{O}_-$ maps to the sector with eigenvalue $-1$ of $\widetilde{F}$ as desired. It is then straightforward to check that

$$\mathcal{O}_-H_{3-} = H_{\text{IRL}}\mathcal{O}_-, \qquad \mathcal{O}_-\mathcal{O}_-^\dagger = 1 - \widetilde{F}, \qquad \mathcal{O}_-^\dagger\mathcal{O}_- = 1 + F_3,$$
$$\text{tr}\left[e^{-\beta H_{\text{IRL}}}\left(1 - \widetilde{F}\right)\right] = \text{tr}\left[e^{-\beta H_{3-}}\left(1 + F_3\right)\right]. \tag{100}$$

The integrable Rydberg-blockade ladder partition function is then given as the sum

$$Z_{\text{IRL}} \equiv \text{tr}\left[e^{-\beta H_{\text{IRL}}}\right] = \text{tr}_{\mathcal{H}_3}\left[e^{-\beta H_3}(1 + F_3)\right] + \text{tr}_{\mathcal{H}_{3-}}\left[e^{-\beta H_{3-}}(1 + F_3)\right]. \tag{101}$$

### 4.4 The integrable Rydberg-blockade ladder and zigzag Ising

Here we complete the square in (64) by utilising the fusion-category construction of topological defects [6, 7]. In any lattice model with weights built from the projectors of a fusion category, there exists a topological defect $\mathcal{D}_a$ for each object $a$ in the category. As proved in [7], its weights obey the defect commutation relations (94), and so the defect-creation operators commute with any Hamiltonian comprised of the projectors. These operators $\mathcal{D}_a$ under multiplication obey the same fusion rules as the corresponding objects.

As detailed above, both the integrable Rydberg-blockade ladder and the zigzag Ising model are constructed from the $su(2)_4$ category, and so possess non-trivial defects for $a = \frac{1}{2}, 1, \frac{3}{2}, 2$ ($\mathcal{D}_0$ is the identity). Moreover, the maps with $a$ half-integer interchange integer and half-integer heights, and so map between $\mathcal{H}_{\text{IRL}}$ and $\mathcal{H}_{\text{zig}}$. Since the latter Hilbert space is comprised entirely of configurations with half-integer heights and the former with integer heights, the defect-creation operators can be split into block-diagonal form

$$\mathcal{D}_{\frac{1}{2}} = \begin{pmatrix} 0 & \mathcal{D} \\ \mathcal{D}^\dagger & 0 \end{pmatrix}, \qquad \mathcal{D}_1 = \begin{pmatrix} \mathcal{S} & 0 \\ 0 & \widetilde{\mathcal{S}} \end{pmatrix}, \tag{102}$$

so that $\mathcal{D} : \mathcal{H}_{\text{IRL}} \to \mathcal{H}_{\text{zig}}$. Thus we have

$$\mathcal{D} H_{\text{IRL}} = H_{\text{zig}} \mathcal{D}, \qquad \mathcal{D}^\dagger H_{\text{zig}} = H_{\text{IRL}} \mathcal{D}^\dagger. \tag{103}$$

The fusion rules of these operators follow immediately from this construction [7], a valuable feature as the products are not as simple as those for the other maps. Here we have

$$\left(\mathcal{D}_{\frac{1}{2}}\right)^2 = 1 + \mathcal{D}_1 \quad \implies \quad \mathcal{D}\mathcal{D}^\dagger = 1 + \mathcal{S}, \quad \mathcal{D}^\dagger\mathcal{D} = 1 + \widetilde{\mathcal{S}}. \tag{104}$$

The maps $\mathcal{S}$ and $\widetilde{\mathcal{S}}$ take a Hilbert space to itself, and (103) and (104) show that they commute with the respective Hamiltonians. They therefore generate symmetries, and we show in section 5 that they are not invertible.

The matrix elements of $\mathcal{D}_{\frac{1}{2}}$ depend on pairs of nearest-neighbour heights, and so can be displayed pictorially as in (92) and (93). Explicit expressions were derived in section 7.5 of [7], giving



where all are invariant under both horizontal and vertical reflections. We continue to use the convention that the height 1 in the Rydberg ladder is labelled by $e$ (for empty rung). Here we do not show the "half-dots" as in the pictures of [7], but our weights include the appropriate factors to make the $\mathcal{D}_{\frac{1}{2}}$ identical to that defined there.

The defect-creation operator $\mathcal{D}$ is nicest to express as a matrix-product operator. Finding it is straightforward, and the workings are displayed in Appendix A. We find that each state in the $(e, +, -)$ basis of $\mathcal{H}_{\text{IRL}}$ maps to a unique one in the $\pm$ basis of $\mathcal{H}_{\text{zig}}$. The matrix product requires three channels, and is

$$\left(\mathcal{D}\right)^{\{h\}}_{\{\tilde{h}\}} = \text{tr}\left(\widetilde{W}^{h_1}_{\tilde{h}_1} \widetilde{W}^{h_2}_{\tilde{h}_2} \cdots \widetilde{W}^{h_L}_{\tilde{h}_L}\right), \tag{106}$$

$$\widetilde{W}^+_e = \begin{pmatrix} 1 & 0 & 0 \\ 0 & 0 & 1 \\ 0 & 0 & 0 \end{pmatrix}, \quad \widetilde{W}^-_e = \begin{pmatrix} 0 & 0 & 1 \\ 1 & 0 & 0 \\ 0 & 0 & 0 \end{pmatrix}, \quad \widetilde{W}^+_+ = \widetilde{W}^-_- = \begin{pmatrix} 0 & 0 & 0 \\ 0 & 0 & 0 \\ 0 & 1 & 0 \end{pmatrix}, \quad \widetilde{W}^+_- = \widetilde{W}^-_+ = 0.$$

It immediately follows that knowing three successive heights $(\tilde{h}_{j-1}, \tilde{h}_j, \tilde{h}_{j+1})$ determines $h_j$:

$$\mathcal{D}: \quad |e \pm e\rangle \rightarrow |\pm\rangle, \qquad \left\{|\pm e\pm\rangle, |eee\rangle\right\} \rightarrow |+\rangle, \qquad \left\{|\pm ee\rangle, |ee\pm\rangle\right\} \rightarrow |-\rangle. \quad (107)$$

This map relates the $\mathbb{Z}_2 \times \mathbb{Z}_2$ charges as $\mathcal{D}\widetilde{F}_{\text{odd}} = F_{\text{odd}}\mathcal{D}$ and likewise for even.

From (103) and (104), it follows that the partition functions of the zigzag Ising model and Rydberg-blockade ladder are related as

$$\text{tr}\left[e^{-\beta H_{\text{IRL}}}\left(1 + \widetilde{\mathcal{S}}\right)\right] = \text{tr}\left[e^{-\beta H_{\text{zig}}}\left(1 + \mathcal{S}\right)\right]. \quad (108)$$

However, deriving more relations between partition functions of the two models directly is rather tricky, as it requires constructing the symmetry sectors non-trivial under $\mathcal{D}$. This non-invertible map is more intricate than the dualities and orbifold used above; as seen in (104), acting twice yields a non-invertible symmetry, either $\mathcal{S}$ or $\widetilde{\mathcal{S}}$. Moreover, deriving such identities requires defining boundary conditions twisted by these symmetries. While the general framework of [7] yields explicit expressions for them in the same form as (105), they are rather unwieldy.

We find an easier path in section 5. Namely, we exploit the fact that the diagram (64) is commutative, a proof we give next in section 4.5. We then easily can relate the spectra of the untwisted Ising zigzag ladder and the 3-state antiferromagnet, as we discuss below in section 5.2. One then can relate the former to the integrable Rydberg ladder by going around the diagram in the other way and so avoid having to use the non-invertible symmetries $\mathcal{S}$ and $\widetilde{\mathcal{S}}$.

## 4.5 Around the square

We prove here that (64) is commutative, i.e. that the maps are related as

$$\mathcal{K}\mathcal{M} = \mathcal{D}\mathcal{O}. \quad (109)$$

The explicit expressions given in (71, 83, 96, 107) give precise definitions of these maps. Knowing three successive spins $s_{j-1}, s_j, s_{j+1}$ of the three-state antiferromagnet in the $ABC$ basis gives under $\mathcal{K}\mathcal{M}$ and $\mathcal{D}\mathcal{O}$ a specific value $h_j = \pm$ in the $\sigma^x$-diagonal basis of the Ising zigzag ladder. Applying this map for each $j$ to each basis element of $\mathcal{H}_3$ yields the image in $\mathcal{H}_{\text{zig}}$. The proof of (109) then simply requires checking that both maps yield the same $h_j$ for each of the twelve allowed configurations of these three spins. Those with the spin $A$ in the center map as

$$
\begin{array}{cccc}
|BAB\rangle \xrightarrow{\mathcal{M}} |\downarrow\uparrow\rangle & |CAC\rangle \xrightarrow{\mathcal{M}} |\uparrow\downarrow\rangle & |BAC\rangle \xrightarrow{\mathcal{M}} |\downarrow\downarrow\rangle & |CAB\rangle \xrightarrow{\mathcal{M}} |\uparrow\uparrow\rangle \\
\mathcal{O}\downarrow \qquad \downarrow\mathcal{K}\ , & \mathcal{O}\downarrow \qquad \downarrow\mathcal{K}\ , & \mathcal{O}\downarrow \qquad \downarrow\mathcal{K}\ , & \mathcal{O}\downarrow \qquad \downarrow\mathcal{K}\ . \\
|e+e\rangle \xrightarrow{\mathcal{D}} |+\rangle & |e+e\rangle \xrightarrow{\mathcal{D}} |+\rangle & |e-e\rangle \xrightarrow{\mathcal{D}} |-\rangle & |e-e\rangle \xrightarrow{\mathcal{D}} |-\rangle
\end{array}
$$

The one subtlety is that if $s_{j\pm 1} = A$, we also need to know $s_{j\pm 2}$ to determine the state at site $j \pm 1$ in the image of $\mathcal{O}$. However, we do know from (96) that this state will be $+$ or $-$, not $e$. This knowledge is enough to determine the image $h_j$ under $\mathcal{D}$, as follows from (107):

$$
\begin{array}{cccc}
|ABC\rangle \xrightarrow{\mathcal{M}} |\uparrow\uparrow\rangle & |ACB\rangle \xrightarrow{\mathcal{M}} |\downarrow\downarrow\rangle & |BCA\rangle \xrightarrow{\mathcal{M}} |\uparrow\uparrow\rangle & |CBA\rangle \xrightarrow{\mathcal{M}} |\downarrow\downarrow\rangle \\
\mathcal{O}\downarrow \qquad \downarrow\mathcal{K}\ , & \mathcal{O}\downarrow \qquad \downarrow\mathcal{K}\ , & \mathcal{O}\downarrow \qquad \downarrow\mathcal{K}\ , & \mathcal{O}\downarrow \qquad \downarrow\mathcal{K}\ . \\
|\pm ee\rangle \xrightarrow{\mathcal{D}} |-\rangle & |\mp ee\rangle \xrightarrow{\mathcal{D}} |-\rangle & |ee\pm\rangle \xrightarrow{\mathcal{D}} |-\rangle & |ee\mp\rangle \xrightarrow{\mathcal{D}} |-\rangle
\end{array}
$$

The remaining relations to be checked are

$$
\begin{array}{cccc}
|ABA\rangle \xrightarrow{\mathcal{M}} |\uparrow\downarrow\rangle & |ACA\rangle \xrightarrow{\mathcal{M}} |\downarrow\uparrow\rangle & |BCB\rangle \xrightarrow{\mathcal{M}} |\uparrow\downarrow\rangle & |CBC\rangle \xrightarrow{\mathcal{M}} |\downarrow\uparrow\rangle \\
\mathcal{O}\downarrow \quad \downarrow\mathcal{K}, & \mathcal{O}\downarrow \quad \downarrow\mathcal{K}, & \mathcal{O}\downarrow \quad \downarrow\mathcal{K}, & \mathcal{O}\downarrow \quad \downarrow\mathcal{K}. \\
|\pm e\pm\rangle \xrightarrow[\mathcal{D}]{} |+\rangle & |\mp e\mp\rangle \xrightarrow[\mathcal{D}]{} |+\rangle & |eee\rangle \xrightarrow[\mathcal{D}]{} |+\rangle & |eee\rangle \xrightarrow[\mathcal{D}]{} |+\rangle
\end{array}
$$

We thus have proved (109).

## 5 Non-invertible symmetries

We have shown how non-invertible mappings relate our four Hamiltonians, giving linear identities between their partition functions. In this section we go further, and describe the non-invertible symmetries of the integrable Rydberg and Ising ladders, as displayed in (64). We derive the remaining identities needed to finish off Table 1.

We first summarise the invertible symmetries and their interrelations. The three-state antiferromagnet has an $S_3$ symmetry generated by $F_3$ and $R$, where $R^3 = 1$ and $(F_3)^2 = 1$ with $F_3 R = R^2 F_3$. Since $\mathcal{M}R = \mathcal{M}$, $R$ does not map to a symmetry generator in XXZ. On the other hand, the definition (83) results in $\mathcal{M}F_3 = F\mathcal{M}$, so $F_3$ is related to the spin-flip symmetry in XXZ. The latter symmetry does *not* map under $\mathcal{K}$ to the Ising zigzag ladder, as $\mathcal{K}F = \mathcal{K}$ follows from (71). However, the $\mathbb{Z}_2 \times \mathbb{Z}_2$ symmetries present at even $L$ in the two ladders are related: $\mathcal{D}\widetilde{F}_{\mathrm{odd}} = F_{\mathrm{odd}}\mathcal{D}$ and likewise for even. The XXZ Hamiltonian (4) has a $U(1)$ symmetry $Q$ with generator defined in (46). The three-state antiferromagnet does as well, as given in (53). The two charges are related by $\mathcal{M}Q_3 = Q\mathcal{M}$.

### 5.1 The $U(1)$ remnants $\mathcal{Q}_{\mathbf{zig}}$ and $\widetilde{\mathcal{Q}}$

The Rydberg and zigzag Ising ladders have no $U(1)$ symmetries. Indeed, one cannot map the XXZ $U(1)$ generator to zigzag Ising:

$$
\mathcal{K}Q\mathcal{K}^{\dagger} = \tfrac{1}{2}\mathcal{K}Q(1+F)\mathcal{K}^{\dagger} = \tfrac{1}{2}\mathcal{K}(1-F)Q\mathcal{K}^{\dagger} = 0, \tag{110}
$$

where we used $\mathcal{K}F = \mathcal{K}$ and $FQ = -QF$. This failure however suggests how a remnant does survive: Although $Q$ cannot be mapped, $Q^2$ can. We therefore define

$$
\mathcal{Q}_{\mathrm{zig}} = \tfrac{1}{2}\mathcal{K}Q^2\mathcal{K}^{\dagger} = \tfrac{1}{2}(1+F_{\mathrm{zig}})\sum_{j=1}^{L}\sum_{k=0}^{L-1}(-1)^{k+1}\prod_{l=0}^{k}\sigma_{j+l}^{x}, \tag{111}
$$

where as always indices are mod $L$. The explicit expression is found by using the commutation relations in (72), and one can check directly that it commutes with $H_{\mathrm{zig}}$. Although it is a relic of the $U(1)$ symmetry of XXZ, it is non-vanishing only in half the Hilbert space, as $F_{\mathrm{zig}}\mathcal{Q}_{\mathrm{zig}} = \mathcal{Q}_{\mathrm{zig}}$. Although the charge is diagonal in the $\pm$ basis, to compute it in practice it is easiest to simply map the state to XXZ, compute the charge there and then square it.

A similar construction for the Rydberg ladder is needed, as $\mathcal{O}Q_3\mathcal{O}^{\dagger} = 0$ as well. However, working out an explicit expression by using $\widetilde{\mathcal{Q}} = \mathcal{O}(Q_3)^2\mathcal{O}^{\dagger}$ is rather difficult. We instead just guess an expression by analogy with (111) and check that it is conserved. It is

$$
\widetilde{\mathcal{Q}} = \tfrac{1}{2}(1+\widetilde{F})\sum_{j=1}^{L}\sum_{k=0}^{L-1}\prod_{l=0}^{k}(-1)^{1-n_{j+l}^{-}}, \tag{112}
$$

where it is useful to recall that $\widetilde{F} = (-1)^{\sum_j (1-n_j^-)}$. This symmetry generator is obviously diagonal in the $(e, +, -)$ basis, and it is easy to check that it commutes with each of the four terms in (59) individually.

The $U(1)$ remnants are not the only non-invertible symmetries: the operators $\mathcal{S}$ and $\widetilde{\mathcal{S}}$ defined in (104) commute with the zigzag Ising and integrable Rydberg ladder Hamiltonians respectively. The latter results in a self-duality that we discuss in section 5.2. Here we show however that the former does not yield anything new, as it can be expressed in terms of $\mathcal{Q}_{\mathrm{zig}}$.

The relation (109) between mappings proves a useful tool, as the maps between zigzag Ising and Rydberg can be rewritten using (98) as

$$\mathcal{KMO}^\dagger = \mathcal{D}\big(1 + \widetilde{F}\big) = \big(1 + F_{\mathrm{zig}}\big)\mathcal{D}\,. \tag{113}$$

Using this expression with $\mathcal{S} = \mathcal{DD}^\dagger - 1$ gives

$$\begin{aligned}
\big(\mathcal{S} + 1\big)\big(1 + F_{\mathrm{zig}}\big) &= \tfrac{1}{2}\mathcal{KMO}^\dagger \mathcal{OM}^\dagger \mathcal{K}^\dagger \\
&= \tfrac{1}{2}\mathcal{KM}\big(1 + F_3\big)\mathcal{M}^\dagger \mathcal{K}^\dagger \\
&= \tfrac{1}{2}\mathcal{KMM}^\dagger \mathcal{K}^\dagger\big(1 + F_{\mathrm{zig}}\big) \\
&= \big(1 + 2\cos\big(\tfrac{2}{3}\pi\sqrt{\mathcal{Q}_{\mathrm{zig}}}\big)\big)\big(1 + F_{\mathrm{zig}}\big)\,.
\end{aligned} \tag{114}$$

To find a simpler expression for $\mathcal{S}$, we utilise the fact that the defect-creation operators obey the same fusion rules as the objects in the category [7]. Here the algebra (34) guarantees there be another symmetry operator $\mathcal{D}_2$. When acting on $\mathcal{H}_{\mathrm{zig}}$ it obeys

$$\mathcal{D}_2\mathcal{S} = \mathcal{SD}_2 = \mathcal{S}\,, \qquad \big(\mathcal{S}\big)^2 = 1 + \mathcal{S} + \mathcal{D}_2\,. \tag{115}$$

By explicit computation using the expressions from [7] or from (107), or by consistency with (114) and (115), one finds that $\mathcal{D}_2 = F_{\mathrm{zig}}$. Then (115) requires that $\mathcal{S}F_{\mathrm{zig}} = F_{\mathrm{zig}}$, and (114) simplifies to

$$\mathcal{S} = \big(1 + F_{\mathrm{zig}}\big)\cos\big(\tfrac{2}{3}\pi\sqrt{\mathcal{Q}_{\mathrm{zig}}}\big)\,. \tag{116}$$

Thus $\mathcal{S}$ does not generate a new symmetry of the Ising zigzag ladder. The expression (116) is in agreement with (115), as

$$\mathcal{S}^2 = \big(1 + F_{\mathrm{zig}}\big)\big(1 + \cos\big(\tfrac{4}{3}\pi\sqrt{\mathcal{Q}_{\mathrm{zig}}}\big)\big) = 1 + \mathcal{S} + F_{\mathrm{zig}}\,, \tag{117}$$

where we used the fact that $\cos\big(2\pi\sqrt{\mathcal{Q}_{\mathrm{zig}}}\big) = 1$.

## 5.2 The non-invertible self-duality $\widetilde{\mathcal{S}}$

The symmetry generator $\widetilde{\mathcal{S}} = \mathcal{D}^\dagger\mathcal{D} - 1$ for the Rydberg ladder does give us something new. A useful expression for it can be derived along the same lines as for $\mathcal{S}$. Using (113) gives

$$\begin{aligned}
\big(\widetilde{\mathcal{S}} + 1\big)\big(1 + \widetilde{F}\big) &= \tfrac{1}{2}\mathcal{OM}^\dagger \mathcal{K}^\dagger \mathcal{KMO}^\dagger = \tfrac{1}{2}\mathcal{OM}^\dagger\big(1 + F\big)\mathcal{MO}^\dagger = \mathcal{OM}^\dagger \mathcal{MO}^\dagger \\
&= \mathcal{O}\big(1 + R + R^2\big)\mathcal{O}^\dagger\,,
\end{aligned} \tag{118}$$

where we recall that $\mathcal{O}F_3 = \mathcal{O}$ and $\mathcal{M}^\dagger F = F_3\mathcal{M}^\dagger$. The category fusion rules also require $\widetilde{\mathcal{S}}\widetilde{F} = \widetilde{\mathcal{S}}$, so

$$\widetilde{\mathcal{S}} = \tfrac{1}{2}\mathcal{O}\big(R + R^2\big)\mathcal{O}^\dagger = \mathcal{ORO}^\dagger\,, \tag{119}$$

where we used $F_3 R^2 = R F_3$ in the last step. Consistency with the fusion rules follows from

$$\big(\widetilde{\mathcal{S}}\big)^2 = \mathcal{O}R\big(1 + F_3\big)R\mathcal{O}^\dagger = 1 + \widetilde{F} + \widetilde{\mathcal{S}}\,. \tag{120}$$

This fusion rule is the analog of the latter of (115) following from the fusion-category construction. It means that $\widetilde{S}$ has three eigenvalues: 2, $-1$, and 0. We finally can complete Table 1 with

$$\mathcal{D}\mathcal{D}^\dagger = 1 + \mathcal{S} = 1 + \left(1 + F_{\text{zig}}\right)\cos\tfrac{2\pi}{3}\sqrt{\mathcal{Q}_{\text{zig}}}\,, \qquad \mathcal{D}^\dagger\mathcal{D} = 1 + \widetilde{S} = 1 + \mathcal{O}R\mathcal{O}^\dagger\,. \tag{121}$$

The operator $\widetilde{S}$ generates a non-invertible symmetry of the integrable Rydberg ladder, as it acts non-trivially only on the sector invariant under $\widetilde{F}$. (The analogous construction using $\mathcal{O}_-$ does not work for the other sector because $R$ does not preserve the modified Hilbert space $\mathcal{H}_{3-}$.) There exists a matrix-product expression for $\widetilde{S}$, but in practice it proves to be much easier to use (119). One maps to the antiferromagnet using $\mathcal{O}^\dagger$ from (96), acts with $\mathbb{Z}_3$ symmetry operators, and then maps back using $\mathcal{O}$. For example, for $L=4$,

$$
\begin{array}{ccc}
|e\,e\,e\,e\rangle & \xrightarrow{\;\mathcal{O}^\dagger\;} & |BCBC\rangle \;+\; |CBCB\rangle \\[2pt]
\widetilde{S}\big\downarrow & & \big\downarrow R \\[2pt]
|e+e+\rangle \;+\; |+e+e\rangle & \xleftarrow{\;\mathcal{O}\;} & |CACA\rangle \;+\; |ACAC\rangle
\end{array}
\tag{122}
$$

with the extension to all even $L$ obvious. When $L$ is odd, however, $\widetilde{S}|\ldots e\,e\,e\ldots\rangle = 0$, as $\mathcal{O}^\dagger$ annihilates it. It is apparent from (122) that $\widetilde{S}$ is not diagonal in the $(e,+,-)$ basis, while $\widetilde{\mathcal{Q}}$ is. The former therefore can not be written in terms of the latter, as opposed to the analogous case (116) in the Ising zigzag ladder. Another example is

$$
\begin{array}{ccc}
|-e\,e\,e\,e\rangle & \xrightarrow{\;\mathcal{O}^\dagger\;} & |ABCBC\rangle \;+\; |ACBCB\rangle \\[2pt]
\widetilde{S}\big\downarrow & & \big\downarrow R \\[2pt]
|e\,e+e-\rangle + |e-e+e\rangle & \xleftarrow{\;\mathcal{O}\;} & |BCACA\rangle \;+\; |BACAC\rangle
\end{array}
\tag{123}
$$

Neither example is obvious, to say the least.

All states with charge $-1$ under the global symmetry $\widetilde{F}$ are annihilated by $\widetilde{S} = \mathcal{D}^\dagger\mathcal{D} - 1$. These states therefore are not annihilated by $\mathcal{D}$, and must be in the image of $\mathcal{D}^\dagger$. All states with charge 1 under $\widetilde{F}$ are in the image of $\mathcal{O}$, as apparent from (98). Thus all eigenstates of $H_{\text{IRL}}$ are in the image of $\mathcal{O}$ or of $\mathcal{D}^\dagger$, with those having eigenvalue 2 under $\widetilde{S}$ in both. These maps thus cover the entire spectrum of $H_{\text{IRL}}$ without need of twisted boundary conditions.

We refer to the symmetry generated by $\widetilde{S}$ as a non-invertible "self-duality", as it mixes the individual terms in (59) like Kramers-Wannier does. Indeed, using (119) we find

$$\widetilde{S}p_j = \left(p_j^\dagger + s_{j-1}s_{j+1}\right)\widetilde{S}, \quad \widetilde{S}p_j^\dagger = \left(p_j + s_{j-1}s_{j+1}\right)\widetilde{S}, \quad \widetilde{S}s_{j-1}s_{j+1} = \left(p_j + p_j^\dagger\right)\widetilde{S}. \tag{124}$$

Thus the operators comprising the Hamiltonian (59)

$$\hat{O}_j^1 = p_j + p_j^\dagger + s_{j-1}s_{j+1}\,, \qquad \hat{O}_j^\Delta = n_j^- + \left(n_{j-1}^e - n_{j+1}^e\right)^2\,, \tag{125}$$

each commute with $\widetilde{S}$. Moreover, certain combinations are odd under this duality. Defining

$$\hat{O}_j^w = 2s_{j-1}s_{j+1} - p_j - p_j^\dagger\,, \qquad \hat{O}_j^t = \left(n_{j-1}^e - n_{j+1}^e\right)^2 - 2n_j^-\,, \tag{126}$$

yields $\widetilde{S}\hat{O}_j^w = -\hat{O}_j^w\widetilde{S}$ and $\widetilde{S}\hat{O}_j^t = -\hat{O}_j^t\widetilde{S}$.

## 5.3 Spontaneous breaking of the self-duality

Since the XXZ chain in equilibrium is well understood [45], we can use non-invertible maps to understand the physics of the other three models. All models are critical for $|\Delta| \leq 1$, and we discuss the corresponding conformal field theories in section 6. At $\Delta = -1$ the "fermi" velocity goes to zero and the dispersion relation of the low-lying excitations is $E \propto k^2$. At $\Delta = 1$, the XXZ chain has an exact $SU(2)$ symmetry and a Kosterlitz-Thouless transition to a gapped phase we discuss here. In particular, we find a nice application of the non-invertible symmetries to the integrable Rydberg ladder with $\Delta > 1$.

We first analyse the ground states in all the models in this region. As $\Delta \to \infty$, the two XXZ ground states at even $L$ become simply $|\uparrow\downarrow\uparrow\downarrow \ldots\rangle$ and $|\downarrow\uparrow\downarrow\uparrow \ldots\rangle$. Since it takes order $L$ actions of $H_{XXZ}$ to mix these two states, the splitting between the ground states must be exponentially small in $\Delta L$ for $\Delta$ large. This exponentially small splitting and hence these two ground states persist until the Kosterlitz-Thouless transition to the gapless phase at $\Delta = 1$. The $\mathbb{Z}_2$ spin-flip symmetry generated by $F$ is therefore spontaneously broken. As the Mermin-Wagner theorem requires, the $U(1)$ symmetry cannot be spontaneously broken, and the ground states are annihilated by $Q$.

For each of the models at $\Delta \to \infty$, the ground states are given by any configuration where $P_j = 0$. In the three-state antiferromagnetic chain, there are six such states for even $L$: $|ABABAB\ldots\rangle$ and its permutations under the $S_3$. These all indeed map under $\mathcal{M}$ to one of the two XXZ ground states. The $S_3$ symmetry is thus spontaneously broken for all $\Delta > 1$. The Ising zigzag ladder, on the other hand, has a unique ground state in this phase, becoming $|++++\ldots\rangle$ as $\Delta \to \infty$. Thus no symmetries are spontaneously broken here.

The integrable Rydberg ladder is the most interesting case. As $\Delta \to \infty$, it has three ground states *not* related by any conventional symmetry:

$$|e\,e\,e\,e\,e\,e\,\ldots\rangle, \quad |+e+e+e\ldots\rangle, \quad |e+e+e+\ldots\rangle.$$

Thus one might expect that for finite $\Delta$, the degeneracy is lifted. Remarkably, the non-invertible self-duality requires that the degeneracy persists. Indeed, from (122) we see that $\widetilde{\mathcal{S}}$ maps between these states while commuting with $H_{IRL}$. Even though we no longer know the exact ground states for $\Delta$ finite, $\widetilde{\mathcal{S}}$ must still map between them until the gap closes. In XXZ the gap does not close until $\Delta = 1$, so there are three ground states for all $\Delta > 1$. Thus in this gapped phase, the non-invertible self-duality is spontaneously broken! In the three-parameter phase diagram discussed in our companion paper [30], we show how this line describes a first-order transition between a $\mathbb{Z}_2$-ordered phase and a disordered phase, similar to what happens in the chain [57].

It is worth contrasting this novel behaviour with the better-known behaviour in the gapped phase $\Delta < -1$. All the models possess exact ground states throughout the phase. The XXZ chain has two exact ferromagnetic ground states, $|\uparrow\uparrow\uparrow\uparrow \ldots\rangle$ and $|\downarrow\downarrow\downarrow\downarrow \ldots\rangle$. Rigorous work gives a gap of $-\Delta-1$ in this region [65]. The corresponding ground state for the Ising zigzag ladder is unique: $|----\ldots\rangle$ in the $\sigma^x$-diagonal basis. It indeed is the image of both XXZ ground states under $\mathcal{K}$ for any $L$. However, the exact ground states of $H_3$ only occur at $L$ a multiple of 3, where $\mathcal{M}^\dagger$ does not annihilate the XXZ ground states. Indeed, these six exact ground states are $|ABCABC\ldots\rangle$ and its permutations under the $S_3$ symmetry. The same restriction applies for $H_{IRL}$, where the three ground states are $|-e\,e-e\,e\ldots\rangle$ and its two translations. The non-invertible symmetry $\widetilde{\mathcal{S}}$ does mix the ground states and so is spontaneously broken as with $\Delta > 1$, but the result is less dramatic here, as translation invariance guarantees the degeneracies. In all four models, these exact ground states are at the extremal values of the respective charges $Q$, $\mathcal{Q}_{zig}$, $Q_3$, and $\widetilde{\mathcal{Q}}$. At the gapless point $\Delta = -1$, the XXZ chain has an $SU(2)$ symmetry for even $L$, and so a $2L+1$-dimensional multiplet of ground states. The other

models also possess more ground states here, with the number depending on $L$ and what of the $SU(2)$ symmetry survive under the mappings.

# 6 CFTs in the continuum

All the work we did in constructing the maps between models pays off in this section. We use these maps and the techniques of conformal field theory to compute the *exact* spectrum of each model in the continuum limit of its critical region. We then go on to make a detailed correspondence between lattice and CFT symmetries.

## 6.1 CFT partition functions

The XXZ chain with $-1 < \Delta \le 1$ has an elegant field-theory description of the continuum limit in terms of a free massless bosonic field $\Phi(x, t)$, where $x$ and $t$ are space and time coordinates [66, 67]. The bosonic field $\Phi$ takes values on a circle, so we identify $\Phi \sim \Phi + 2\pi r$ for a positive radius $r$. In this free-boson field theory, the dimensions of all the operators can be computed exactly. The most convenient way to give the results is in terms of the partition function for spacetime a torus. It can be computed exactly either directly from the path integral or by using the powerful tools of conformal field theory (CFT). We give a brief overview of how this works in Appendix B. Following the conventions of [68], the free-boson CFT partition function on the torus is

$$\overline{Z}(r) = \frac{1}{\eta^2} \sum_{l,m=-\infty}^{\infty} q^{\left(\frac{l}{2r}\right)^2 + (mr)^2}, \tag{127}$$

where the Dedekind eta function is $\eta = q^{\frac{1}{24}} \prod_{n=1}^{\infty}(1 - q^n)$. We explain shortly how the parameter $q$ (not to be confused with the quantum-group parameter mentioned above) depends on physical quantities. The integer $m$ is the $U(1)$ charge of the corresponding configuration, which is the winding number of the boson. The invariance $\overline{Z}(r) = \overline{Z}(\frac{1}{2r})$ can be thought of as electric-magnetic duality. The scaling dimensions come from expanding $q^{\frac{1}{12}}\overline{Z}(r)$ in a series in $q$; the exponent then gives the dimension of the operator creating the corresponding state [69, 70].

The precise identification of the continuum limit of $H_{\mathrm{XXZ}}$ with the field theory has long been known [66, 67]. Comparing the scaling dimensions to the results coming from integrability [45] gives an exact relation between the boson radius $r$ and the XXZ coupling $\Delta$:

$$Z_{\mathrm{XXZ}} \to \overline{Z}(r), \qquad \text{where } \Delta = -\cos(2\pi r^2), \quad 0 < r \le \frac{1}{\sqrt{2}}. \tag{128}$$

To match the partition functions precisely, the parameter $q$ in (127) depends on the system size $L$, the inverse temperature $\beta$, and the XXZ fermi velocity. The $\Delta$-dependence of the latter is known exactly from integrability, giving

$$q = e^{2\pi v_F \frac{\beta}{L}}, \qquad v_F = \frac{\sin(2\pi r^2)}{1 - 2r^2} = \pi \frac{\sqrt{1 - \Delta^2}}{\arccos \Delta}. \tag{129}$$

By now (128) and (129) have been thoroughly and convincingly established.

The field-theory limits of the lattice generators $Q$ and $F$ are easy to find. The $U(1)$ charge of the field theory is $\overline{Q} \propto \int_0^L \frac{d}{dx}\Phi = \Phi(L) - \Phi(0)$, and we normalize it so that its eigenvalue is the winding number $m$. To relate it to the lattice generator $Q$, consider $\Delta = 1$ and $r = \frac{1}{\sqrt{2}}$, where the symmetry is enhanced to $SU(2)$. (The corresponding CFT is the $SU(2)_1$ WZW model [71].)

Here the four states with $(l, m) = (\pm 1, 0)$ and $(0, \pm 1)$ form a triplet and a singlet under the $SU(2)$ and so the raising and lowering operators have charge 1. However, the lattice $SU(2)$ raising operator $\sum_j S_j^+$ changes $Q$ by 2 in our normalisation, as it flips a down spin to an up one. Thus we find that in the continuum limit $Q \to 2\overline{Q}$. The lattice spin-flip generator $F$ anticommutes with $Q$, so its continuum limit must anticommute with $\overline{Q}$. Thus $F \to \overline{F}$, where $\overline{F} : \Phi \to -\Phi$.

Because the maps we have constructed are exact on the lattice, they remain exact in the field theory. Any eigenstate of the Hamiltonian in one model maps to an eigenstate of the other with the same energy. However, because the mappings are non-invertible, the partition functions are not identical. Instead, as described in section 4, they obey linear relations that involve inserting symmetry generators and allowing for twisted boundary conditions.

The CFT partition functions we study all can be defined in terms of orbifolds [31] built from a finite abelian symmetry group $G$ of dimension $|G|$. Orbifolding means effectively to gauge or "mod out" by the symmetry. Thus one projects onto a subspace invariant under the symmetry. However, modding out by the symmetry also allows new states in the theory, corresponding to those with twisted boundary conditions. The orbifold partition function is

$$\overline{Z}_G(r) = \frac{1}{|G|} \sum_{g,h \in G} \overline{Z}_{g,h}(r), \quad \text{where } \overline{Z}_{g,h}(r) = \text{tr}_{\mathcal{H}_h}\Big[ g e^{-\beta H_h(r)} \Big], \tag{130}$$

with $H_h(r)$ the field-theory Hamiltonian at radius $r$ with an $h$-twisted boundary condition, and $\mathcal{H}_h$ the Hilbert space on which it acts. Twisted boundary conditions and orbifolds of a free bosonic field are well understood; for a complete discussion see [32]. The $h$-twisted boundary condition is natural to describe in two-dimensional spacetime: it amounts to placing a defect wrapping around the Euclidean "time" direction, just as the operator insertion of $g$ corresponds to a defect across the spatial direction. The same goes on the lattice [7].

The linear relation (130) between the partition functions of a model and its orbifold are fairly obviously continuum versions of lattice relations such as (90). To make precise contact between the two, we need to rephrase our results in the language of defect boundary conditions. We already did so for the three-state antiferromagnet at the end of section 4.3. There we defined the $F_3$-twisted Hamiltonian $H_{3-}$ by using the usual Hamiltonian with the caveat that for terms in the Hamiltonian involving spins at sites $L$ and 1, we must utilise $s_{L+j} = \bar{s}_j$. Since the bar means to exchange the roles of $B$ and $C$, we have inserted an $F_3$ defect between sites $L$ and 1, as the approach of [7] makes precise. Similarly, the twisted XXZ Hamiltonian $H_-$ defined in (79) can be interpreted as arising from an $F$-twisted boundary condition. Namely, we define $\bar{\uparrow} = \downarrow$ and $\bar{\downarrow} = \uparrow$, i.e. the bar is the action of $F$. Using the usual definitions of $S_L$ and $P_L$ with the caveat that the spins involved are taken to be $|s_{L-\frac{1}{2}} \bar{s}_{\frac{1}{2}}\rangle$ (or equivalently $|\bar{s}_{L-\frac{1}{2}} s_{\frac{1}{2}}\rangle$) gives precisely the extra signs in (79) as compared to (4). Finally, the other modified XXZ Hamiltonian $H_\omega$ defined in (89) can be interpreted as boundary conditions twisted by the $\mathbb{Z}_3$ symmetry generated by $\omega^Q$. Twisting by it amounts simply to acting on one of the spins in $S_L$ and $P_L$ with it, resulting in (89).

The continuum partition functions for all our models follow, as now we can identify the linear relations between partition functions as various orbifolds. The identity (81) giving $Z_{\text{zig}}$ as a sum of XXZ partition functions is an orbifold by the spin-flip symmetry $F$. Indeed, both maps $\mathcal{K}$ and $\mathcal{K}_-$ are non-trivial only in the $F$-even sector, hence the $(1 + F)$ in (81), which in the continuum turns into the $(1 + \overline{F})$ in (130). Moreover, $\mathcal{K}_-$ requires utilising the $F$-twisted Hamiltonian (79), again just as in (130). Thus the four terms in (81) become in the continuum limit (130) with $G$ the $\mathbb{Z}_2$ generated by $\overline{F} : \Phi \to -\Phi$. The resulting partition function describes states even under $\overline{F}$ as well twisted ones with $\Phi(x + L, t) = -\Phi(x, t)$. It has

been computed [31,32], giving

$$Z_{\text{zig}} \;\to\; \overline{Z}_{\overline{F}}(r) = \tfrac{1}{2}\Big(\overline{Z}(r) + 2\overline{Z}\big(\tfrac{1}{2\sqrt{2}}\big) - \overline{Z}\big(\tfrac{1}{\sqrt{2}}\big)\Big), \tag{131}$$

in the continuum limit. Here and everywhere $r$ is related to $\Delta$ via (128). The lattice Kramers-Wannier duality implemented by $\mathcal{K}$ and $\mathcal{K}_-$ thus amounts to a $\mathbb{Z}_2$ orbifold by $\overline{F}$. The partition function (131) also describes the continuum limit of the Ashkin-Teller model [72], which also can be reached from XXZ by a lattice orbifold [33].

We thus have in (131) an exact expression for all the scaling dimensions in the continuum limit of the integrable Ising zigzag ladder for $-1 < \Delta \le 1$. The factor of $\tfrac{1}{2}$ in front of $\overline{Z}(r)$ results from throwing out states odd under $\overline{F}$, while the remaining terms describe the twisted sectors. The lowest-dimension operators in the latter have dimension $\tfrac{1}{8}$ and $\tfrac{9}{8}$ for all $r$. The $r$-independence arises because identifying $\Phi$ with $-\Phi$ is possible only with winding number zero. Even though the $\mathbb{Z}_2$ symmetry $\overline{F}$ is no longer present, a new $\mathbb{Z}_2$ symmetry $\overline{F}_{\text{zig}}$ arises, under which all states in the twisted sector are odd. One then can orbifold by $\overline{F}_{\text{zig}}$ and recover the original free boson [68]. It is natural then to have the lattice symmetry $F_{\text{zig}} \to \overline{F}_{\text{zig}}$ in the continuum limit. Thus the map $\mathcal{K}^\dagger$ implements a lattice orbifold by $F_{\text{zig}}$.

A nice check on (131) comes at $\Delta = 1$, where the $U(1)$ symmetry of XXZ is enhanced to $SU(2)$. The orbifold by $F$ breaks this symmetry, but a $U(1)$ subgroup survives: $\mathcal{K}Q_x = Q_{zz}\mathcal{K}$, where

$$Q_x = \sum_{j=1}^{L} X_{j+\frac{1}{2}}, \qquad Q_{zz} = \sum_{j=1}^{L} \sigma_j^z \sigma_{j+1}^z \quad \implies \quad \big[Q_{zz}, H_{\text{zig}}\big] = 0, \;\; \text{for } \Delta = 1. \tag{132}$$

Since the Ising zigzag ladder at $\Delta = 1$ has a $U(1)$ symmetry with local generators, its partition function should be that of a free boson, and indeed from (131) we see that $\overline{Z}_{\overline{F}}\big(\tfrac{1}{\sqrt{2}}\big) = \overline{Z}\big(\tfrac{1}{2\sqrt{2}}\big)$.

The partition functions of the other two models are also related in such a fashion. The identity (91) with $a = 0$ gives the partition function for the three-state antiferromagnet. It arises from the orbifold of XXZ by its $\mathbb{Z}_3$ symmetry generated by $\omega^Q$: The partition functions $Z_\omega$ and $Z_{\omega^2}$ are those for the two kinds of twisted boundary conditions, while the sum over $\nu$ projects onto the sector invariant under $\omega^Q$. Since $\omega^Q \to \omega^{-\overline{Q}}$ in the continuum, the CFT partition function from (130) must be that of the free boson orbifolded with $G = \mathbb{Z}_3$. Recalling that the eigenvalue of $\overline{Q}$ is the winding number $m$, projecting on the $\mathbb{Z}_3$-invariant subspace amounts to requiring that $m$ be a multiple of 3, just as required to make $\mathcal{M}^\dagger$ non-trivial. As the winding number obeys $\Phi(L) - \Phi(0) = 2\pi m r$, this restriction is equivalent to sending $r \to 3r$. The twisted sectors correspond to the presence of operators $e^{i\frac{m}{3r}\Phi}$, again consistent with sending $r \to 3r$. The partition function of the three-state antiferromagnet thus becomes

$$Z_3 \;\to\; \overline{Z}_{\mathbb{Z}_3}(r) = \overline{Z}(3r). \tag{133}$$

This identification agrees with that already made [48–50] for $\Delta = -\tfrac{1}{2}$, describing the Hamiltonian limit of the square-lattice zero-temperature antiferromagnet [21]. The corresponding radius $r = 6^{-1/2}$ is the unique value where $\overline{Z}(3r) = \overline{Z}(r)$.

The maps $\mathcal{O}$ and $\mathcal{O}_-$ from the three-state antiferromagnet to the integrable Rydberg ladder were already described in section 4.3 in terms of a lattice orbifold by the symmetry $F_3$. The resulting sum for $Z_{\text{IRL}}$ in (101) is precisely a sum of the form (130) with $G = \mathbb{Z}_2$ generated by $F_3$. Since we know the continuum limit of the $Z_3$ from (133), the Rydberg CFT partition function follows:

$$Z_{\text{IRL}} \;\to\; \overline{Z}_{\overline{F}_3}(r) = \tfrac{1}{2}\Big(\overline{Z}(3r) + 2\overline{Z}\big(\tfrac{1}{2\sqrt{2}}\big) - \overline{Z}\big(\tfrac{1}{\sqrt{2}}\big)\Big), \tag{134}$$

in harmony with the numerics of [25] and the analytics of [23]. This expression has a number of interesting physical consequences for the Rydberg-blockade ladder at and near its critical line. We explore them in our companion paper [30], and also match lattice operators to their CFT counterparts.

The orbifolds described in this section can be summarised as

$$Z_{\text{zig}} \xleftarrow{F} Z_{\text{XXZ}} \xrightarrow{\omega^Q} Z_3 \xrightarrow{F_3} Z_{\text{IRL}}, \tag{135}$$

where the generator of the discrete group $G$ is given. Abelian orbifolds can all be reversed, because a new discrete symmetry appears to replace the one that has been gauged. For example, if one orbifolds the Ising zigzag ladder by its $\mathbb{Z}_2$ symmetry generated by $F_{\text{zig}}$, one recovers the XXZ chain. Orbifolding the three-state antiferromagnet by $R$ gives XXZ as well. These symmetries for the reverse orbifolds can be read off from the products of non-invertible maps in Table 1, giving

$$Z_{\text{zig}} \xrightarrow{F_{\text{zig}}} Z_{\text{XXZ}} \xleftarrow{R} Z_3 \xleftarrow{\widetilde{F}} Z_{\text{IRL}}. \tag{136}$$

The generators $F$ and $\omega^Q$ form a $S_3$ symmetry group of XXZ. Since $\mathcal{K}F = F_3\mathcal{K}$, (135) indicates that the integrable Rydberg ladder is an $S_3$ orbifold, where one first orbifolds by the $\mathbb{Z}_3$ and then the $\mathbb{Z}_2$. Reversing the order (i.e. going around (64) the other way) seems trickier. One starts with the $\mathbb{Z}_2$ generated by $F$ to get to $Z_{\text{zig}}$. Then one expects that the next step is to orbifold by $\mathbb{Z}_3$ generated by $\omega^{\sqrt{Q}}$, as suggested by (121), but it is not clear to what extent such an orbifold is precisely defined.

## 6.2 Symmetries in the orbifold CFT

The discrete and $U(1)$ symmetries of the XXZ and antiferromagnetic chains are easy to understand in the CFT. The lattice and continuum correspondences between discrete and non-invertible symmetries in the other two models are not so obvious, and we explain them here.

The CFTs for the zigzag Ising and Rydberg ladders are both orbifolds under $\overline{F} : \Phi \to -\Phi$ (and $\theta \to -\theta$). The orbifold has two key effects. One is that only $\overline{F}$-invariant combinations appear in the theory. In zig-zag Ising, using the free-boson operators defined in (B.2) gives $C_{l,m} = \frac{1}{2}(V_{l,m} + V_{-l,-m})$ for all non-negative $l, m$. For Rydberg, one must define $\widetilde{V}_{l,m}$ of dimension $\widetilde{x}_{l,m}$ in the free-boson CFT with radius $\widetilde{r} = 3r$ to give

$$\widetilde{C}_{l,m} = \tfrac{1}{2}\big(\widetilde{V}_{l,m} + \widetilde{V}_{-l,-m}\big) = \cos\left(\tfrac{l}{\widetilde{r}}\Phi + 2m\theta\widetilde{r}\right), \qquad \widetilde{x}_{l.m} = \tfrac{l^2}{4\widetilde{r}^2} + m^2\widetilde{r}^2. \tag{137}$$

The other distinction of orbifold theories is the presence of twisted fields, which arise because field configurations with $\Phi(x + L, t) = -\Phi(x, t)$ are now allowed. There turn out to be two pairs of such fields [31] denoted $\sigma_{1,2}$ and $\tau_{1,2}$, of $r$-independent scaling dimensions $x = \frac{1}{8}$ and $\frac{9}{8}$ respectively.

Requiring that operators be of the form (137) breaks the $U(1)$ symmetry of the free-boson CFT. The discrete symmetry of the free-boson orbifold CFT at generic $r$ turns out to be the dihedral group $D_4$, just as in the zig-zag Ising ladder. The best way to gain intuition is to treat it as two coupled Ising CFTs. At $r = \frac{1}{2}$, the two decouple, and the orbifold partition function (131) is the square of the Ising model one [68, 72]. Indeed, for $L$ even the Hamiltonian (63) at the corresponding value $\Delta = 0$ decouples into two commuting pieces $H_{\text{zig}} = H_1 + H_2$, where

$$H_1 = \frac{1}{2}\sum_{j \text{ even}}\left(\sigma^z_{j-1}\sigma^x_j\sigma^z_{j+1} + \sigma^z_j\sigma^z_{j+2}\right), \qquad H_2 = \frac{1}{2}\sum_{j \text{ odd}}\left(\sigma^z_{j-1}\sigma^x_j\sigma^z_{j+1} + \sigma^z_j\sigma^z_{j+2}\right). \tag{138}$$

The commuting Hamiltonians $H_1$ and $H_2$ individually are equivalent to Ising, a fact easiest to see by writing the operators in terms of Majorana fermions.[1] Including terms $-\Delta \sigma_j^x$ couples the two models and changes the radius $r$ of the CFT. The individual Ising $\mathbb{Z}_2$ flip symmetries $F_{\text{even}}$ and $F_{\text{odd}}$ are preserved when the two theories are coupled, as is the $\mathbb{Z}_2$ symmetry $E$ exchanging the two copies. The corresponding continuum limits $\overline{F}_1$, $\overline{F}_2$ and $\overline{E}$ generate the $D_4$ of the CFT.

The decoupling into two Ising models at $r = \frac{1}{2}$ allows us to identify the symmetry properties of the CFT fields directly. The two twist fields $\sigma_1$, $\sigma_2$ of the CFT are the continuum limits of the two Ising spin fields, so that each is odd under the corresponding Ising flip symmetry $\overline{F}_1$ and $\overline{F}_2$. The operator $C_{0,1}$ is the product $\sigma_1 \sigma_2$, indeed of dimension $\frac{1}{4}$ at $r = \frac{1}{2}$. It thus is odd under both the $\overline{F}_a$, but even under $\overline{E}$. Each Ising model has an "energy" operator $\epsilon_a$ of dimension 1 found in the operator product expansion $\sigma_a \cdot \sigma_a \sim 1 + \epsilon_a$. (We write operator products in fusion-algebra form, where we omit coefficients and only include primary fields.) Since $C_{0,1} \cdot C_{0,1} \sim 1 + C_{0,2}$, we identify $C_{0,2}$ with $\epsilon_1 + \epsilon_2$. It indeed is of dimension 1 at $r = \frac{1}{2}$, as is $C_{1,0}$. The latter therefore must be the difference of the Ising energy fields. Thus $C_{0,2}$ is invariant under the full $D_4$, while $C_{1,0}$ is invariant under both $\overline{F}_a$ but odd under $\overline{E}$. The dihedral generators in the CFT therefore act as

$$
\begin{aligned}
\overline{F}_a : \quad & \sigma_b \to (-1)^{a+b-1} \sigma_b, \quad \tau_b \to (-1)^{a+b-1} \tau_b, \quad C_{l,m} \to (-1)^m C_{l,m}, \\
\overline{E} : \quad & \sigma_b \to \sigma_{3-b}, \quad \tau_b \to \tau_{3-b}, \quad C_{l,m} \to (-1)^l C_{l,m}.
\end{aligned}
\tag{139}
$$

Whereas the dimensions of the $C_{l,m}$ change when $r$ varies, the form of the operator-product expansion stays the same. The symmetries (139) must therefore apply for all $r$. Since the CFT for the integrable Rydberg ladder is obtained by sending $r \to \widetilde{r} = 3r$, the operators $\widetilde{C}_{l,m}$ must have the same discrete symmetry as the corresponding $C_{l,m}$ in zigzag Ising.

The CFT operators do not in general have a definite charge under the non-invertible symmetries $\widetilde{S}$ and $\widetilde{Q}$. However, we can understand which operators are invariant under them. In the XXZ chain/free-boson CFT, we have identified the eigenvalue of $Q \to 2\overline{Q}$ with the winding number $2m$. The Rydberg non-invertible symmetry charge $\widetilde{Q}$ from (112) relates to the map of $(Q_3)^2$, which in turn relates to $Q^2$. Thus a state with $Q$ eigenvalue $2m$ and even under $F$ maps to a state with eigenvalue $4m^2$ under $\widetilde{Q}$. The operator $\widetilde{C}_{l,m}$ from (137) maps to $C_{l,m}$ in the XXZ chain, which changes winding number by $\pm m$ there. Thus only the $\widetilde{C}_{l,0}$ commute with $\widetilde{Q}$.

We now can understand the $D_4$ symmetry in the Rydberg ladder. The lattice symmetry generated by $\widetilde{F}_{\text{even}}$ and $\widetilde{F}_{\text{odd}}$ must correspond to the $\mathbb{Z}_2 \times \mathbb{Z}_2$ subgroup generated by $\overline{F}_1$ and $\overline{F}_2$. The lattice version of $\overline{E}$ is not as obvious. However, the unit cell used to take the continuum limit here is four sites, so translation by a single site turns into an internal symmetry of the CFT [73]. Moreover, the lattice translation generator obeys $T\widetilde{F}_{\text{even}} = \widetilde{F}_{\text{odd}}T$, akin to $\overline{E}\,\overline{F}_1 = \overline{F}_2\overline{E}$. We thus identify $\overline{F}_1\overline{E}$ as the continuum consequence of lattice translation symmetry, as $(\overline{F}_1\overline{E})^4 = 1$. The CFT symmetries (139) then indicate that the state created by $\widetilde{C}_{l,m}$ in the CFT must have eigenvalue $(-1)^{l+m}$ under lattice translation, consistent with the numerical results displayed in [30].

The self-duality $\widetilde{S}$ is related to the $\mathbb{Z}_3$ charge $R$ of the antiferromagnet, as derived in (119). Furthermore, as described in section (4.3), sectors not invariant under $R$ map to twisted boundary conditions in the XXZ chain. The continuum version of this statement is that sectors in the antiferromagnet with non-trivial electric charges $l \bmod 3 \neq 0$ only map to XXZ with twisted boundary conditions, as they do not exist in XXZ (recall that the radius $\widetilde{r} = 3r$). Therefore only operators $\widetilde{C}_{l,m}$ from (137) with $l \bmod 3 = 0$ commute with $\widetilde{S}$.

---

[1] Take $\psi_{2j} = i\sigma_1^x \sigma_3^x \cdots \sigma_{j-1}^x \sigma_{j-1}^z \sigma_j^z$ for even $j$ and $\psi_{2j} = i\sigma_2^x \sigma_4^x \cdots \sigma_{j-1}^x \sigma_{j-1}^z \sigma_j^z$ for odd $j$, along with $\psi_{2j+1} = i\psi_{2j}\sigma_{j-1}^z \sigma_{j+1}^z$.

# 7 Conclusions

We have explored in detail four quantum Hamiltonians and the non-invertible mappings between them, as summarised in (64). The mappings obey a rich set of interrelations involving both conventional and non-invertible symmetries, summarised in Table 1. Our results illustrate that Kramers-Wannier duality is not the only non-invertible mapping between lattice models with interesting consequences both formal and physical. The algebraic approach arising from (2) enabled us to make precise the connections between four one-parameter families of integrable models, and provided a valuable tool in developing the non-invertible maps between them as well. We are optimistic that further investigation will result in further progress, both formally and practically.

Since the XXZ chain has been intensively studied for decades, we utilised the non-invertible mappings and symmetries to relate its physics to the other models. While the gross features are the same (a gapless phase for $|\Delta| \leq 1$, gapped for $|\Delta| > 1$), some of the phenomena are quite different. One striking example is for $\Delta > 1$. In XXZ, the phase is the standard antiferromagnetic one, with spontaneously broken spin-flip symmetry. Similarly, the three-state antiferromagnet has spontaneously broken $S_3$ symmetry. In the Ising zigzag ladder, no symmetries are spontaneously broken. The Rydberg-blockade ladder, however, here is characterised by three ground states related only by the non-invertible symmetry $\widetilde{S}$, which thus is spontaneously broken.

Fusion categories played an important role in our analysis. Although using them to define the mappings may seem an elaborate formalism at first glance, we hope our results show that categories provide a tool fairly straightforward to apply. Our results, in particular the fact that the diagram (64) commutes, strongly suggest that utilising further the general approach of [20] will result in even more progress.

The seldom-exploited structure of the venerable XXZ chain we have utilised might have practical implications for XXZ itself. Its Hamiltonian can be written in terms of generators of an algebra not involving the coupling $\Delta$, as opposed to the usual Temperley-Lieb and quantum-group-algebra approaches. So whereas this property misses all the intricacies of representation theory depending on $q$ being a root of unity, the presentation here in terms of trivalent graphs is elegant and simple. One interesting issue to address is the conserved quantities. There is an elegant and explicit expression for them in terms of Temperley-Lieb generators [74], so it is natural to wonder if one could be built in terms of our generators.

The integrable Rydberg-blockade ladder seems very interesting in its own right. Our work provides exact results useful for tackling the difficult general problem. We have made a step in our companion paper [30], where we find a variety of interesting ordered phases once the couplings are varied away from the integrable line. For example, the $\Delta > 1$ line, with three ground states and a spontaneously broken non-invertible symmetry, is a first-order transition line similar to that of the Rydberg-blockade chain [57]. However, some aspects are very different. For example, while our $c = 1$ critical line is analogous to the Ising critical line in the Rydberg-blockade chain, here it separates the disordered region from a phase with spontaneously broken $D_4$ dihedral symmetry, not just a $\mathbb{Z}_2$. Moreover, since the lattice model possesses two non-invertible symmetries, the orbifold CFT describing the continuum limit of this line must as well. While we found the fields invariant under them, we do not understand their full action. We leave this interesting issue to future work.

## Acknowledgments

P.F. thanks Slava Krushkal for many recent and long-ago conversations on chromatic algebras, and Hosho Katsura and Yuan Miao for helpful comments on the manuscript.

**Funding information** This work has been supported by the EPSRC Grant no. EP/S020527/1.

## A  Matrix-product form of $\mathcal{D}_{\frac{1}{2}}$

To express the defect-creation operator $\mathcal{D}_{\frac{1}{2}}$ in terms of matrix products we organise the weights in (105) into the matrix

$$
W = \frac{1}{\sqrt{2}}
\begin{pmatrix}
0 & 2^{\frac{1}{4}} & 2^{\frac{1}{4}} & 0 \\
2^{\frac{1}{4}} & 1 & -1 & 2^{\frac{1}{4}} \\
2^{\frac{1}{4}} & -1 & 1 & 2^{\frac{1}{4}} \\
0 & 2^{\frac{1}{4}} & 2^{\frac{1}{4}} & 0
\end{pmatrix},
\tag{A.1}
$$

where each row and column are labeled by a pair of heights across the defect, namely $(\tilde{h}_j, h_j) = (0, \frac{1}{2}), (e, \frac{1}{2}), (e, \frac{3}{2}), (2, \frac{3}{2})$. One should think of $W$ as propagating the defect weights down the chain. The trick to get the MPO is to characterise each of the four pairs of heights in terms of a diagonal $4 \times 4$ matrix, namely $\mathcal{P}_0^{\frac{1}{2}} = \mathrm{diag}(1, 0, 0, 0)$, $\mathcal{P}_e^{\frac{1}{2}} = \mathrm{diag}(0, 1, 0, 0)$, etc. The matrix elements of the defect operators $\mathcal{D}$ and $\mathcal{D}^{\dagger}$ then are

$$
\left(\mathcal{D}_{\frac{1}{2}}\right)^{\{h\}}_{\{\tilde{h}\}} = \mathrm{tr}\left(\mathcal{P}^{h_1}_{\tilde{h}_1} W \mathcal{P}^{h_2}_{\tilde{h}_2} W \cdots \mathcal{P}^{h_L}_{\tilde{h}_L} W\right),
\tag{A.2}
$$

where the trace is in the $4 \times 4$ matrix space.

To rewrite these matrix elements in a more transparent form, we utilise the eigenvalues and eigenvectors of $W$, which are

$$
\lambda_1 = 2^{\frac{3}{4}}, \qquad \lambda_2 = -2^{-\frac{3}{4}}, \qquad \lambda_3 = \sqrt{2}, \qquad \lambda_0 = 0,
$$

$$
v_1 = \frac{1}{2}\begin{pmatrix} 1 \\ 1 \\ 1 \\ 1 \end{pmatrix}, \quad
v_2 = \frac{1}{2}\begin{pmatrix} 1 \\ -1 \\ -1 \\ 1 \end{pmatrix}, \quad
v_3 = \frac{1}{\sqrt{2}}\begin{pmatrix} 0 \\ 1 \\ -1 \\ 0 \end{pmatrix}, \quad
v_0 = \frac{1}{2}\begin{pmatrix} 1 \\ 0 \\ 0 \\ -1 \end{pmatrix}.
\tag{A.3}
$$

so that

$$
W = \sum_{r=0,1,2,3} \lambda_r v_r v_r^T = \sum_{r=1,2,3} \lambda_r v_r v_r^T,
\tag{A.4}
$$

$$
\begin{aligned}
\mathcal{P}_0^{\frac{1}{2}} &= \mathrm{diag}(1,0,0,0) = \tfrac{1}{4}\left(v_1 + v_2 + \sqrt{2}v_0\right)\left(v_1^T + v_2^T + \sqrt{2}v_0^T\right), \\
\mathcal{P}_2^{\frac{3}{2}} &= \mathrm{diag}(0,0,0,1) = \tfrac{1}{4}\left(v_1 + v_2 - \sqrt{2}v_0\right)\left(v_1^T + v_2^T - \sqrt{2}v_0^T\right), \\
\mathcal{P}_e^+ &= \mathrm{diag}(0, \tfrac{1}{\sqrt{2}}, \tfrac{1}{\sqrt{2}}, 0) = \tfrac{1}{\sqrt{2}}\left(\tfrac{1}{2}(v_1 - v_2)(v_1^T - v_2^T) + v_3 v_3^T\right) = \tfrac{1}{\sqrt{2}}\left(v_- v_-^T + v_3 v_3^T\right), \\
\mathcal{P}_e^- &= \mathrm{diag}(0, \tfrac{1}{\sqrt{2}}, -\tfrac{1}{\sqrt{2}}, 0) = \tfrac{1}{2}\left(v_3\left(v_1^T - v_2^T\right) + (v_1 - v_2)v_3^T\right) = \tfrac{1}{\sqrt{2}}\left(v_3 v_-^T + v_- v_3^T\right),
\end{aligned}
\tag{A.5}
$$

where $v_{\pm} = (v_1 \pm v_2)/\sqrt{2}$ so that $W v_{\pm} = \lambda_1 v_{\mp}$, The latter two relations use the $\pm$ basis of $H_{\mathrm{zig}}$, where states are labelled by the eigenvalues of $\sigma_j^x$. Since $\lambda_0 = 0$, the eigenvector $v_0$ does not

contribute to the defect matrix elements and thus we can omit any term with a $\nu_0$ from the $\mathcal{P}$ in (106). Thus the first two projectors make the same contribution to the matrix element, so that we can take linear combinations of them and use the $\pm$ states for both $\mathcal{H}_{\text{IRL}}$ and $\mathcal{H}_{\text{zig}}$. Thus using

$$\mathcal{P}_+^+ = \mathcal{P}_-^- = \tfrac{1}{2}\left(\nu_+\nu_+^T\right), \qquad \mathcal{P}_+^- = \mathcal{P}_-^+ = 0, \tag{A.6}$$

in (A.2) gives the correct matrix elements in these bases.

Using (A.5) and the replacement (A.6) in (A.2) yields a rather simple matrix-product operator in the $\pm$ bases. It is simple to check that all non-vanishing matrix elements are 1. The result is given in (106), where the three channels correspond to $\nu_3$, $\nu_+$ and $\nu_-$, where we have omitted the zero-eigenvalue channel because it does not contribute to the matrix elements, We have also absorbed $W$ into the $\mathcal{P}$, which means the form (106) is left-right asymmetric. The resulting operator remains symmetric, however, as clear from (107). Indeed, for $\tilde{h}_j = +$ and $\tilde{h}_j = -$ in the IRL, acting with $\mathcal{D}_{\frac{1}{2}}$ automatically yields $h_j = +$ and $h_j = -$ in zigzag Ising respectively. The state $\tilde{h}_j = e$ can result in either $h_j = \pm$, which one being determined by the values of $\tilde{h}_{j-1}$ and $\tilde{h}_{j+1}$ as in (107).

## B The XXZ chain and conformal field theory

As explained in classic work [66,67], the continuum limit of the XXZ chain in the region $-1 < \Delta \leq 1$ is described by a free boson. An intuitive way of understanding why is to define a lattice operator $\phi_j$ at each site $j = 0, 1, \ldots L$ such that $\phi_{j+1} - \phi_j = Z_{j+\frac{1}{2}}$. One then hopes that in the continuum limit, a suitably coarse-grained version of $\phi_j$ renormalizes into a free real bosonic field $\Phi(x,t)$ where $x$ and $t$ are space and time coordinates. The lattice $U(1)$ symmetry charge is by construction $Q = \phi_L - \phi_0$, so the field-theory $U(1)$ charge must be $\overline{Q} \propto \int_0^L \partial_x \Phi = \Phi(L) - \Phi(0)$. Thus even though we have periodic boundary conditions in the XXZ chain, the operator $\phi_j$ and the field $\Phi(x,t)$ cannot be periodic in the presence of configurations with non-zero charge. Instead we must "compactify" $\Phi(x,t)$ to take values on a circle, i.e. identify the field values $\Phi(x) \sim \Phi(x) + 2\pi r$ for some radius $r$. The periodic boundary conditions in XXZ then allow for having field configurations where $\Phi(x + L) = \Phi(x) + 2\pi m r$, where the charge $m$ is an integer called the winding number.

The bosonic field-theory action in Euclidean spacetime describing the XXZ chain is [66,67, 75]

$$S = -\frac{1}{2\pi} \int_0^\beta d\tau \int_0^L dx \left( \left|\left(\partial_x + i\partial_\tau\right)\Phi(x,i\tau)\right|^2 + \kappa \cos\left(\tfrac{2}{r}\Phi\right) \right). \tag{B.1}$$

Spacetime is a torus of size $L \times \beta$, with $\beta$ the inverse temperature. For $\kappa = 0$, this action is conformally invariant, and so we can avail ourselves of the powerful methods of conformal field theory. In particular, the bosonic field can be split into chiral and antichiral components (right and left-moving in real time) $\Phi = \varphi_R(x + i\tau) + \varphi_L(x - i\tau)$, and then define a dual field as $\theta = \varphi_R - \varphi_L$. Operators in this conformal field theory of dimension $x_{l,m}$ are then defined as

$$V_{l,m} = e^{i\frac{l}{r}\Phi + 2imr\theta}, \qquad x_{l,m} = \frac{l^2}{4r^2} + m^2 r^2, \tag{B.2}$$

where the electric charge $l$ and the magnetic charge/winding number $m$ need to be integers for consistency with $\Phi \sim \Phi + 2\pi r$. Standard techniques [68] give the torus partition function to be that given in (127).

The last term in (B.1) arises because the allowed values of $\phi_j$ are discrete and equally spaced. A potential $\kappa \cos(a\Phi)$ is indeed minimized when $\Phi$ is at discrete equally spaced values. The subtle part is to determine the factor $a$. As apparent from (B.2), the lowest-dimension

operators of this sort present in the free-boson theory are $V_{\pm 1,0}$, not the $V_{\pm 2,0}$ in (B.1). There are (at least) three ways of understanding why the former operators do not appear. A precise but not very intuitive method is to utilise the exact results coming from the integrability of the XXZ (and XYZ) chains [45]. Computing the exact dimensions and comparing to those from (B.2) yields the relation (128) between $\Delta$ and $r$ and the factor of 2 in (B.1) [66, 67] (We note in passing that while the relationship is not rigorous, it is exact; the two concepts are orthogonal.) A more intuitive explanation comes from the fact that at $\Delta = 1$, the XXZ interaction is antiferromagnetic, so the unit cell used for the coarse-graining procedure should be two sites in order for the Néel state to correspond to $\Phi = 0$ [73]. With a two-site unit cell, the exact translation symmetry of the lattice model results in an internal symmetry $\Phi \to \Phi + \pi r$ of the free-boson theory. The operators $V_{\pm 1,0}$ are not invariant under this symmetry and so cannot be added to the action, making (B.1) the simplest action consistent with all the symmetries. A third way of understanding the factor of 2 comes from a careful examination of the anomalies, which result in an "emanant" symmetry of the field theory forbidding $V_{\pm 1,0}$ from appearing in the action [76].

Since the final term in (B.1) is irrelevant for $0 < r < 1/\sqrt{2}$, $\kappa$ renormalizes to zero when coarse-graining. We thus can ignore it in this region, so the XXZ chain with $-1 < \Delta \leq 1$ is described by the free-boson CFT. The KT transition at $\Delta = 1$ corresponds to $V_{\pm 2,0}$ becoming relevant, and so for $\Delta > 1$ the CFT no longer applies. The CFT description also breaks down as $\Delta \to -1$, but for a very different reason. Again the integrability can be used to compute the Fermi velocity as a function of $\Delta$, yielding (129). As $\Delta \to -1$, $r \to 0$, and so $v_F$ also goes to zero. At $\Delta = -1$, the model remains gapless, but the dispersion relation is quadratic.

The CFT has two $U(1)$ symmetries. The lattice $U(1)$ symmetry corresponds to $\theta \to \theta + \text{const}$, which indeed has charge $\propto \int dx\, \partial_t \theta = \int dx\, \partial_x \Phi$ as explained above. The second one corresponds to $\Phi \to \Phi + \text{const}$, and is present only in the critical region where $\kappa$ renormalises to zero. It is not present in the lattice model, only in the CFT, and so is emergent.

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
