# Peer review of "From the XXZ chain to the integrable Rydberg-blockade ladder via non-invertible duality defects"

_SciPost Physics, doi:SciPost Phys. 16, 127 (2024)_

## Round 3 · Referee Report · Linnea Grans-Samuelsson (Referee 1) · 2023-10-20

Strengths
-
Provides a useful, unified setting for dealing with four different lattice models, and goes into a good amount of detail about the mappings between them. Also exploits the mappings to find interesting properties of the models.
-
Pedagogical and clear.
-
Diagram 4.1 and table 1 nicely summarize the main results about non-invertible mappings and symmetries, and provide an excellent introduction to section 4. More generally, the paper is well structured.
Weaknesses
- (Minor point) Much care is taken in Section 2 to show graphical presentations of the algebra defined in eqs (2.1) and (2.2), and fusion categories are also introduced from the perspective of labelled graphs. However, in latter sections there is very little graphical interpretation given (even though topological defects often do lend themselves well to a graphical description), making the paper feel a bit disconnected between the first and second parts.
Report
The authors show that apart from the XXZ spin chain, three other models have Hamiltonians that can be written in terms of this other algebra: the 3-state antiferromagnet, the integrable Rydberg-blockade ladder and the Ising zigzag ladder. As they are described by the same algebra, there must be relations between these models; detailing and exploiting these relations is the main theme of the paper. The authors take care to repeatedly remind the reader of the difference from the Temperley-Lieb approach, such as the parameter $\Delta$ no longer appearing in the algebra. This is helpful, as the Temperley-Lieb approach may be very familiar to readers of this paper.
The authors carefully work out non-invertible mappings and symmetries between and within the four models using the formalism of topological defects. These mappings are first helpfully summarized in diagram 4.1 and table 1, giving the reader a convenient overview, after which each mapping is presented in detail. These mappings are used to relate the partition functions to each other, and to derive properties of the models for different regimes of $\Delta$. By using in particular the relation between the XXZ spin chain (whose CFT description in the critical regime is very well understood) and the other three models, the exact CFT spectrum of the other three models in the critical regime is computed. The framework of orbifolding used for the CFT partition functions is pedagogically presented.
I recommend that this paper is published in SciPost Physics after minor revisions (see requested changes). The models under consideration are interesting models, and these mappings between and within them are very useful.
Requested changes
-
Each of subsections 4.1, 4.2 and 4.3 finish with a relation between the partition functions of the models discussed in that subsection. However, in subsection 4.4 there is no such final expression between the integrable Rydberg-blockade ladder and the zigzag Ising ladder, leaving the reader hanging. If the ultimate goal is to always express partition functions in terms of $Z_{XXZ}$ (as described in the introduction), and the authors do not intend to spell out any relation directly between $Z_{IRL}$ and $Z_{zig}$, that should be (re)stated explicitly both at the very end of subsection 4.4 and around the beginning of section 4, to remind the reader of this. This would increase the clarity of section 4.
-
(Minor fix) In the heading of subsection 5.1, the first $\mathcal{Q}$ should have a subscript "zig" -- especially since there's another $Q$ that is otherwise only distinguished by font choice.
-
(Optional suggestion) At the beginning of subsection 4.3, a graphical way of presenting the non-invertible maps and the defect commutation relations is shown, and it is noted that it can be used for all maps that are discussed. Perhaps it could be shown in some form already at the beginning of section 4?

---

## Round 3 · Referee Report · Anonymous (Referee 2) · 2024-1-26

Strengths
- uncovers an algebra underlying four different integrable spin chains
- carefully written, self-contained
- there is no doubt that the paper is correct
Weaknesses
- a clearer separation between new models, new methods, new results on the one hand, and review of existing literature on the other hand, would be nice
Report
They introduce the Baxterization of that algebra in eq. (3.4).
Then they show that this algebra occurs in four different quantum spin chains: the XXZ spin chain ---intringuingly, for any value of the anisotropy $\Delta$, not related to $q$ here---, the 3-state Potts chain, and two RSOS-type chains (`Rydberg blockade' chain and Ising ladder).
Then they exploit this algebraic understanding to carefully analyze mappings between these four different models, both in their lattice formulation (chapters 4 and 5) and in the continuum limit (chapter 6).
The paper is very carefully written. I think it could be published as it is. My only criticism is that, because the paper is self-contained, it lacks conciseness. It contains a lot of standard material, which is not always clearly separated from the new results. For some readers, it might be difficult to appreciate what is new here. If the authors could clarify this, for instance by re-organizing slightly sections. 2 and 3, the manuscript would improve.

---

## Round 3 · Referee Report · Anonymous (Referee 3) · 2024-1-30

Strengths
- Interesting
- Detailed
- Well written
Report
Report on the paper:
From the XXZ chain to the integrable Rydberg-blockade ladder via non-invertible duality defects
Let me start my report by giving my overall assessment of the paper. The novelty, interest and quality of this paper is dual to speed with which this report was produced. This paper clearly satisfies the criteria for publication in SciPost easily, and so I recommend this paper for publication in SciPost. Below, I raise some issues/questions (marked with an *), which the authors can act on, if they deem this appropriate.
The paper studies four integrable models, starting from the XXZ model. They do this by means of an algebraic structure different from the usually employed Temperley-Lieb algebra. This different algebraic structure allows the authors study non-invertible duality/topological defect maps between these models, which are closely related to modular tensor categories, as for instance utilised in anyon chains.
In Section 2, the authors introduce and study the algebra underlying this paper. In particular, they introduce two sets of generators, S_j and P_j, in terms of which the hamiltonians are given.
- The operators P_j act like projector operators, as is clear from the second relation in eq. (2.2). For the P_j to be projectors, they should also be hermitian. However, the authors do not commend on the hermiticity of the P_j. It seems that the authors implicitly assume this, because otherwise the hamiltonian obtained would not be hermitian. On the other hand, non-hermitian 'hamiltonians' can be very interesting. Does the construction work for non-hermitian P_j (and/or S_j)?
It is appreciated that the authors explain the graphical presentation of the algebra(s) in detail in section 2.2 and 2.3.
In section 2.4, the authors relate the algebraic structure they utilise to fusion categories, in particular, they use a specific set of F-symbols.
-
The F-symbols have a large amount of gauge freedom. Using a different gauge can typically change the appearance of the operators/hamiltonian, but does not typically lead to different physics. Is this also the case here?
-
A related question, would a (essentially) different solution of the pentagon equation lead to different hamiltonians (provided more than one unitary solution of the pentagon equation exist)?
In section 3, the authors derive four different hamiltonians using their algebraic structure.
-
What is the purpose of the sentence following eq. (3.5)?
-
In the sentence following eq. (3.6) it is stated that the YBE requires that T(u) and T(u') to commute, but doesn't this follow if R satisfies the YBE? So, should 'requires' be replaced by 'guarantees' ?
-
After eq. 3.14, it would be useful to remind the reader that S_j corresponds to the first term in 3.14, P_j to the second.
With the hamiltonians at hand, the authors study the non-invertible mappings between them in great detail. For instance, they derive the products of non-invertible maps as given in Table 1, as well as the relation between the partition functions, such as eq. 4.18.
- The relation between the partition functions of the Rydberg ladder and Ising zigzag chain is not stated in in section 4.4. Perhaps the authors can comment on this omission?
In section 5, the authors study the non-invertible symmetries, and utilise them to study the physics of the models, by relating them to the well-studied XXZ chain.
- In this context, it is clear that no conventional symmetry relates the three ground states of the Rydberg ladder for large \Delta, as the authors state. However, two of these ground states are related by translational symmetry. Is it possible to utilise this in some way?
Finally, in section 6, the authors use their results to obtain the spectra of their models (at criticality) in the continuum limit. This leads to intriguing relations between the partition functions of the models in terms of various orbifolds, 'in both directions'.
- As a final comment/question, I am wondering if the authors could comment on the applicability of their algebraic approach in different contexts. Is it use restricted to the (type of) models studied in the current (and accompanying) paper, or could this method be applied to other integrable systems, such as S=1 (or even higher spin) integrable chains, or perhaps super-integrable models?
Requested changes
No requested changes, but see the the questions marked by an * in my report.

---

## Round 4 · Referee Report · Anonymous (Referee 3) · 2024-3-15

Report

In their resubmission, the authors dealt with my remarks in an appropriate way. I also think that they responded in a satisfactory way to the other reports. I therefor recommend the paper for publication in Scipost Physics.

---

## Round 4 · Referee Report · Anonymous (Referee 2) · 2024-3-15

Report

I am satisfied with the author's reply. I understand that they already put a lot of effort into making the paper self-contained, and don't want to re-organize it at this stage. The manuscript can be published as it is.

---

## Round 4 · Referee Report · Linnea Grans-Samuelsson (Referee 1) · 2024-3-22

Report

The authors address the questions and suggestions by me and the other referees well in their resubmission. I recommend this paper for publication in Scipost Physics.

---

## Round 4 · Author Response

We thank the referees for their comments. Rather than reply to the reports individually, we reply here, including the comments for clarity (since there are many).

Report 3: Questions: 1. The operators P_j act like projector operators, as is clear from the second relation in eq. (2.2). For the P_j to be projectors, they should also be hermitian. However, the authors do not commend on the hermiticity of the P_j. It seems that the authors implicitly assume this, because otherwise the hamiltonian obtained would not be hermitian. On the other hand, non-hermitian 'hamiltonians' can be very interesting. Does the construction work for non-hermitian P_j (and/or S_j)?

We never assumed hermiticity. However, given the intimate connection to the XXZ and other hermitian Hamiltonians, it would be rather surprising if a non-hermitian presentation of (2.2) exists. In particular, the generators would then satisfy more relations if S\ne S^\dagger, which probably would kill the whole thing. We don’t have a proof, however. More plausible is that a generalisation of (2.2) exists with non-Hermitian generators, e.g. SS^\dagger S^\dagger S = 1-P. We don’t know of a systematic approach to the latter problem, however.

  1. The F-symbols have a large amount of gauge freedom. Using a different gauge can typically change the appearance of the operators/hamiltonian, but does not typically lead to different physics. Is this also the case here?

We expect so. Changing the gauge of the su(2)_4 F-symbols from which our Hamiltonians are built should lead to unitarily equivalent Hamiltonians which have the same spectrum.

  1. A related question, would a (essentially) different solution of the pentagon equation lead to different hamiltonians (provided more than one unitary solution of the pentagon equation exist)?

We focus on the su(2)_4 fusion category in the paper, for which all solutions of the pentagon equation should be related by the aforementioned gauge transformations. However, the authors of our Ref 7 did examine the question of whether different categories with the same fusion algebra (e.g. SU(2)_2 and Ising) result in different lattice models via this construction. They checked a variety of examples and never found anything new. However, they did not find a general proof.

  1. What is the purpose of the sentence following eq. (3.5)? [“The second and third Reidemeister moves follow from (3.5) and (3.1) by taking Bj ∝ Rj (i∞)”]

This sentence isn’t essential, but we put it in there for those who like to think about knot polynomials. (In this part of Section 3, it is checked that the braid generators defined as Bj ∝ Rj (i∞), with Rj as defined in Eq. 3.4, satisfy the braid group relations. The second Reidemeister move as stated in Eq. 2.24 is satisfied because Rj(u) satisfies Eq. 3.5. The third Reidemeister move as stated in Eq. 2.25 is satisfied since Rj(u) satisfies the YBE equation 3.1. )

  1. In the sentence following eq. (3.6) it is stated that the YBE requires that T(u) and T(u') to commute, but doesn't this follow if R satisfies the YBE? So, should 'requires' be replaced by 'guarantees' ?

We agree, and replaced ‘requires’ by ‘guarantees’.

  1. After eq. 3.14, it would be useful to remind the reader that S_j corresponds to the first term in 3.14, P_j to the second.

Sure, we added this.

7.The relation between the partition functions of the Rydberg ladder and Ising zigzag chain is not stated in in section 4.4. Perhaps the authors can comment on this omission?

Thanks to the referee for noticing. We added the simplest relation between the two models, which is now equation 4.45. Other relations are rather tricky to obtain directly, and we have now added an explanation as to why in the corresponding paragraph. We have also explained how to avoid these difficulties in the following paragraph using the analysis later in the paper (in and around equation 5.14).

  1. In this context, it is clear that no conventional symmetry relates the three ground states of the Rydberg ladder for large \Delta, as the authors state. However, two of these ground states are related by translational symmetry. Is it possible to utilise this in some way?

Two of the three ground states are indeed related by translation symmetry, and so are expected to be degenerate (up to exponentially small corrections). However, no conventional argument shows that the third ground state is also degenerate with the former two for finite \Delta > 1. That’s where the power of the non-invertible symmetry is particularly impressive.

  1. As a final comment/question, I am wondering if the authors could comment on the applicability of their algebraic approach in different contexts. Is it use restricted to the (type of) models studied in the current (and accompanying) paper, or could this method be applied to other integrable systems, such as S=1 (or even higher spin) integrable chains, or perhaps super-integrable models?

Ref 7 gives a fairly general construction of such models and the corresponding non-invertible symmetries, and other papers (e.g. those by Lootens et al) push this construction even further. Thus our analysis most certainly can be generalised further. However, we focused on this case because we found it a beautiful non-trivial example of this structure where everything could be written out in a fairly simple-to-parse fashion. (Our model is in essence a spin 1 model, the simplest one not writable as a spin-½ model.) It would be great if more general cases could be written out in such a simple fashion, but we will leave that to other authors!

Report 2: Weakness: My only criticism is that, because the paper is self-contained, it lacks conciseness. It contains a lot of standard material, which is not always clearly separated from the new results. For some readers, it might be difficult to appreciate what is new here. If the authors could clarify this, for instance by re-organizing slightly sections. 2 and 3, the manuscript would improve.

Much (perhaps all) of the material in sections 2 and 3 was known before, and we made a substantial effort to provide the correct references. However, the results are scattered throughout the literature, and the connections are far from obvious. We spent considerable time writing those sections, as we were trying to present the material in the clearest and most unified fashion. We don’t know how to start to re-organising them, and we’re not sure how it would improve the manuscript. We think the payoff of writing sections 2 and 3 in a unified fashion makes the applications derived in the remaining sections clear and approachable. We believe it is fair to say that in the earlier treatments it often takes considerable overhead to start extracting applications (and one of us was an author of Ref 7). Our goal here was to reduce this overhead, i.e. to compress it into two hopefully clear sections.

Report 1: Weakness: (Minor weakness) Much care is taken in Section 2 to show graphical presentations of the algebra defined in eqs (2.1) and (2.2), and fusion categories are also introduced from the perspective of labelled graphs. However, in latter sections there is very little graphical interpretation given (even though topological defects often do lend themselves well to a graphical description), making the paper feel a bit disconnected between the first and second parts.

The graphs were used to prove the equivalence between various models, and we drew many of them to make the connections more pedagogically transparent (as we indicated in our reply to Report 2). We then used these equivalences to derive the various results of the later sections, and so didn’t feel the need to repeat these graphs.

Requested changes: 1. Each of subsections 4.1, 4.2 and 4.3 finish with a relation between the partition functions of the models discussed in that subsection. However, in subsection 4.4 there is no such final expression between the integrable Rydberg-blockade ladder and the zigzag Ising ladder, leaving the reader hanging. If the ultimate goal is to always express partition functions in terms of Z_{XXZ} (as described in the introduction), and the authors do not intend to spell out any relation directly between Z_{IRL} and Z_{zig} that should be (re)stated explicitly both at the very end of subsection 4.4 and around the beginning of section 4, to remind the reader of this. This would increase the clarity of section 4.

This comment is similar to that made in Report 3, comment 7, and we thank this referee as well for noticing. We hope our reply there suffices to explain this issue, and how we addressed it.

(Minor fix) In the heading of subsection 5.1, the first Q should have a subscript "zig" -- especially since there's another Q that is otherwise only distinguished by font choice.

Thanks for noticing, we fixed that.

(Optional suggestion) At the beginning of subsection 4.3, a graphical way of presenting the non-invertible maps and the defect commutation relations is shown, and it is noted that it can be used for all maps that are discussed. Perhaps it could be shown in some form already at the beginning of section 4?

Thanks for the suggestion. We had originally pondered doing so. The reason we did not is that it’s a little subtle, and we worried that discussing it in section 4.1 would confuse matters. Namely, the correct and clearest way to do Kramers-Wannier duality involves changing sublattices, as described in detail in our reference 5. Thus the rectangles we draw in section 4.3 have non-trivial spins only at two of the corners in the cases of section 4.1 and 4.2. We thus thought (and still think) it would be better to avoid this issue, as our setup didn’t require explaining this subtlety. So apologies, but we’d rather leave the ordering as is.

---

## Round 4 · List of Changes

The largest one was the two paragraphs added at the end of section 4.4, including equation 4.45, to explain why we didn't supply as simple expressions there as we had in the previous sections. The others were all very minor, and described in our reply to the referees.

---

## Editorial Decision

published